# GraphPlanner: Graph Memory-Augmented Agentic Routing for Multi-Agent LLMs

**Tao Feng, Haozhen Zhang, Zijie Lei, Peixuan Han, Jiaxuan You**
Department of Computer Science
University of Illinois Urbana Champaign Urbana, IL, USA
{taofeng2, jiaxuan}@illinois.edu

## ABSTRACT

LLM routing has achieved promising results in integrating the strengths of diverse models while balancing efficiency and performance. However, to support more realistic and challenging applications, routing must extend into *agentic LLM settings*—where task planning, multi-round cooperation among heterogeneous agents, and memory utilization are indispensable. To address this gap, we propose `GraphPlanner`, a heterogeneous graph memory-augmented agentic router for multi-agent LLMs that generates routing workflows for each query and supports both inductive and transductive inference. `GraphPlanner` formulates workflow generation as a Markov Decision Process (MDP), where at each step it selects both the LLM backbone and the agent role (Planner, Executor, Summarizer). By leveraging a heterogeneous graph, denoted as `GARNet`, to capture interaction memories among queries, agents, and responses, `GraphPlanner` integrates historical memory and workflow memory into richer state representations. The entire pipeline is optimized with reinforcement learning, jointly improving task-specific performance and computational efficiency. We evaluate `GraphPlanner` across 14 diverse LLM tasks and demonstrate that: (1) `GraphPlanner` outperforms strong single- and multi-round routers, improving accuracy by up to 9.3% while reducing GPU cost from 186.26 GiB to 1.04 GiB; (2) `GraphPlanner` generalizes robustly to unseen tasks and LLMs, exhibiting strong zero-shot capabilities; and (3) `GraphPlanner` effectively leverages historical memories, supporting both inductive and transductive inference for more adaptive routing. Our code for `GraphPlanner` is released at https://github.com/ulab-uiuc/GraphPlanner.

## 1 INTRODUCTION

Routing among multi-agent large language models (LLMs) has become a key approach for integrating the strengths of diverse models while balancing efficiency and performance (Shnitzer et al., 2023; Hu et al., 2024a; Chen et al., 2024a; Feng et al., 2024; 2025). Despite this importance, most existing routing methods remain confined to simplified or static settings, which limits their applicability in solving complex real-world tasks (Feng et al., 2025). In contrast, the recent rise of agentic LLMs has shown how multi-agent collaboration can enhance planning, strengthen reasoning, and boost overall performance on complex tasks (Wang et al., 2024a; Qian et al., 2024; Guo et al., 2024; Wu et al., 2024; Barachini & Stary, 2022; Tran et al., 2025). These agentic capabilities highlight the need to revisit routing in more realistic and challenging scenarios, where heterogeneous LLMs differ in capability, cost, and reliability. In such contexts, effective routing is not only beneficial but necessary to fully unlock the potential of agentic LLM systems. Therefore, our paper aims to raise attention to this pressing research question: *How can we extend routers to agentic LLM settings?*

Existing routing approaches fall into single-round and multi-round routers as shown in Table 1. Single-round routers (Shnitzer et al., 2023; Hu et al., 2024a; Chen et al., 2024a; Feng et al., 2024) make one-shot assignments based on query embeddings or classifiers. While simple and efficient, this paradigm lacks the ability to reason over multiple steps, decompose tasks, or coordinate across different LLMs, which limits its effectiveness on complex queries. Multi-round routers (Zhang et al., 2025; Shao et al., 2025) extend flexibility by interleaving reasoning and routing over multiple calls.

Table 1: **Comparison of `GraphPlanner` with existing LLM routers across four dimensions: workflow type, historical memory usage, graph utilization, and model size.** Unlike existing routers, `GraphPlanner` is a lightweight LLM router based on an agentic workflow, which leverages heterogeneous graphs to handle historical memories and thereby facilitate better routing.

| LLM Router | Workflow type | Historical memory usage | Graph utilization | Model size |
|---|---|---|---|---|
| RouterDC (Chen et al., 2024a) | Single-round | ✗ | ✗ | Medium |
| GraphRouter (Feng et al., 2024) | Single-round | ✓ | ✓ | Small |
| R2-Reasoner (Shao et al., 2025) | Multi-round | ✗ | ✗ | Medium |
| Router-R1 (Zhang et al., 2025) | Multi-round | ✗ | ✗ | Large |
| GraphPlanner | Multi-agent | ✓ | ✓ | Small |

However, they do not explicitly model collaboration between LLMs, treating each call as independent rather than part of a cooperative workflow, which leads to redundant calls, context conflicts, and limited use of complementary strengths. Additional related works can be found in Appendix A.

To address these limitations, we generalize routing as a multi-agent coordination problem, where the router must decide not only which LLM backbone to invoke but also which agent role to activate at each step. This shift is crucial because agentic LLM routers can explicitly model specialization and cooperation across multiple agents, turning independent calls into structured workflows. Yet, building an effective agentic LLM router is far from trivial and comes with several challenges. *First, the relations among queries, responses, and LLM candidates are highly diverse and complex in agentic settings.* Unlike single-step assignments, agentic workflows require reasoning over evolving contexts where queries may branch, responses interact, and different models contribute complementary but sometimes conflicting information. Designing a router that can capture and leverage these heterogeneous dependencies is a non-trivial task. *Second, agentic routing involves deferred rewards. Early routing decisions often have long-term effects on the overall outcome, meaning that immediate feedback is insufficient.* For example, an early misallocation may cascade into redundant calls or degraded reasoning quality downstream. This creates a challenging credit assignment problem, requiring the router to balance short-term efficiency with long-term performance. *Third, it remains an open question how to fully exploit abundant historical memories from agentic LLM systems.* Rich traces of past multi-agent workflows contain valuable insights into successful collaboration patterns, error modes, and efficient division of labor. Yet, existing routers rarely make systematic use of this information, leaving a gap in leveraging historical data for improving future coordination.

To tackle the above challenges, we propose `GraphPlanner`, a heterogeneous graph memory-augmented agentic router for multi-agent LLMs that generates multi-agent routing workflows for each query and supports both inductive and transductive inference. Specifically, `GraphPlanner` casts the generation of agentic routing workflows as graph generation within a Markov Decision Process (MDP) (Garcia & Rachelson, 2013). At each step of graph generation, `GraphPlanner` must decide not only which LLM backbone to invoke but also which agent role to activate based on the current state. Without loss of generality, we define the agent profiles as Planner, Executor, and Summarizer, which capture the essential roles in agentic workflows (Barachini & Stary, 2022; Tran et al., 2025). Further, `GraphPlanner` utilizes a heterogeneous graph, denoted as `GARNet`, to model the memories among LLM agents, queries, and responses. By capturing such heterogeneous information, it can fully exploit abundant historical memories as well as the current workflow memories, thereby constructing richer and more informative state representations. Finally, we introduce a deep reinforcement learning algorithm named Proximal Policy Optimization (PPO) (Schulman et al., 2017) into the entire pipeline to jointly optimize task-specific performance of the final answers as well as the associated computational cost.

We evaluate `GraphPlanner` in two phases across 14 tasks spanning 6 domains. In Phase 1, agentic routing is optimized within existing workflows, while Phase 2 focuses on generating workflows for complex agentic tasks. Across both phases, `GraphPlanner` consistently outperforms single-round and multi-round routers, improving average accuracy by +3.8% in Phase 1 and +9.3% in Phase 2, while reducing GPU cost from 186.26 GiB to 1.04 GiB and remaining on the Pareto frontier. Furthermore, `GraphPlanner` demonstrates strong generalization, achieving 78% average accuracy on unseen tasks (20–40% higher than previous routers) and robustly handling unseen LLMs without additional fine-tuning. Finally, by modeling historical memories alongside current workflow memories through `GARNet`, `GraphPlanner` significantly enhances routing decisions and supports both inductive and transductive inference: the inductive mode offers greater efficiency, while the transductive mode yields stronger performance at higher cost.

Figure 1: **Comparison between the agentic router, the single-round router, and the multi-round router.** Specifically, the single-round router selects a model based only on the query, the multi-round router makes sequential selections using accumulated context, and the agentic router leverages a workflow graph to jointly choose agent roles and models for collaborative reasoning. The agentic router enables explicit collaboration and task decomposition by leveraging a workflow memory graph, allowing multiple LLMs with different roles to coordinate more effectively than single/multi-round routers.

## 2 PRELIMINARIES

Routing among multiple large language models (LLMs) has emerged as a crucial paradigm for balancing performance and efficiency. Existing approaches can be broadly categorized into *single-round routers* and *multi-round routers*. Before presenting our formulation of agentic routing, we first review these two settings and highlight their inherent limitations.

**Single-round routers**. In the standard setting, a router takes a text query $q \in \mathcal{Q}$ and directly assigns it to one model from a backbone pool $\mathcal{M} = \{M_1, \ldots, M_K\}$. Formally, a single-round router (Shnitzer et al., 2023; Hu et al., 2024a; Chen et al., 2024a; Feng et al., 2024) $R_{\text{single}}$ as shown in the top-left part of Figure 1 is defined as:

$$m = R_{\text{single}}(q), \quad o = M_m(q), \tag{1}$$

where $m$ denotes the selected model and $o$ is the output generated by $M_m$. This paradigm is simple and efficient, but it lacks the ability to reason, decompose tasks, or coordinate multiple LLMs. As a result, it struggles when facing complex queries that require collaboration across specialized models.

**Multi-round routers**. To improve flexibility, the multi-round router (Zhang et al., 2025; Shao et al., 2025) as shown in the top-right part of Figure 1 considers routing decisions that take into account historical context information. Given a query $q_t$, the router adaptively chooses a backbone model based on both the current query and the context $c_t$, where $c_t$ contains all previous queries, model selections, and outputs from the interaction history:

$$m_t = R_{\text{multi}}(c_t, q_t), \quad o_t = M_{m_t}(q_t). \tag{2}$$

This contextual design enables the router to make more informed decisions by learning from past interactions and model performances. However, this sequential design may still incur redundant calls, risk semantic conflicts in accumulated context, and lack explicit mechanisms for coordinating complementary strengths of different models.

**Agentic routers**. To overcome these limitations, we generalize routing as a *multi-agent coordination problem*. Instead of only selecting a backbone model, the router must also decide which *agent role* (e.g., Planner, Executor, Summarizer) to activate. Given the query $q_t$ and the evolving workflow memory graph $\mathcal{G}_{workflow}$, the agentic router $R_{\text{agentic}}$ as shown in the bottom part of Figure 1 selects:

$$(a_t, m_t) = R_{\text{agentic}}(q_t, \mathcal{G}_{workflow}), \tag{3}$$

where $a_t$ indexes the chosen agent role $A_{a_t}$ and $m_t$ indexes the backbone $M_{m_t}$. The pair $(A_{a_t}, M_{m_t})$ executes on the sub-query, producing intermediate output $o_t$. These outputs are integrated through the workflow and summarized at the final stage to produce the answer. By explicitly modeling agent roles and workflows, agentic routers enable structured collaboration between LLMs, supporting decomposition, multi-role cooperation, and more adaptive decision-making.

## 3 GRAPHPLANNER: GRAPH-BASED AGENTIC LLM ROUTING

As shown in Figure 2, GraphPlanner formulates LLM routing as a sequential decision-making process over agentic workflows. At each step, the router selects both an agent role (planner, executor,

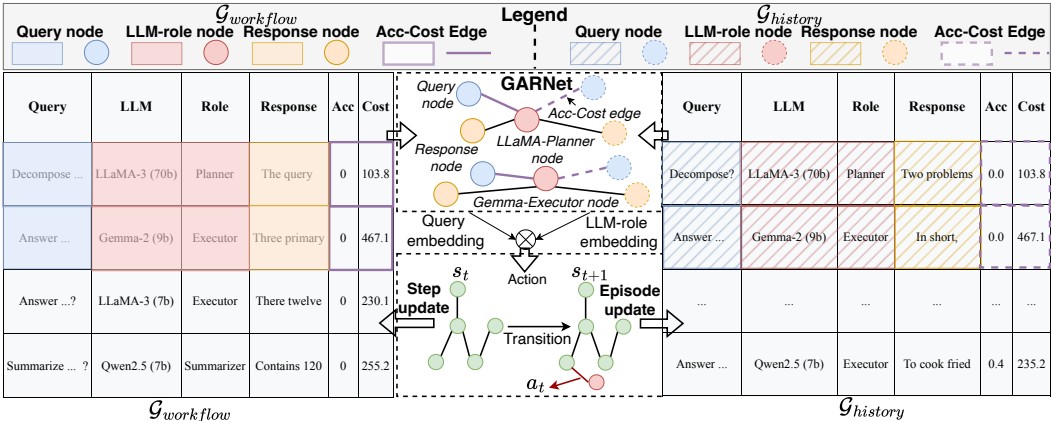

Figure 2: **Overview of `GraphPlanner`.** In `GraphPlanner`, each decision step is guided by `GARNet`, which integrates $\mathcal{G}_{workflow}$ and $\mathcal{G}_{history}$ to produce an action that specifies both the LLM and the agent role. The resulting trajectories are incrementally incorporated into $\mathcal{G}_{workflow}$ at each step, while the complete episode trajectory is consolidated into $\mathcal{G}_{history}$ at the end of the episode. Note that boxes and circles sharing the same color denote a direct mapping relationship.

or summarizer) and an LLM backbone, guided by GARNet which integrates the current workflow memory graph $\mathcal{G}_{workflow}$ and the historical memory graph $\mathcal{G}_{history}$. This graph-based formulation enables context-aware routing and supports end-to-end optimization through RL.

## 3.1 AGENTIC ROUTING WORKFLOW GENERATION AS MARKOV DECISION PROCESS

We cast the agentic routing workflow generation as a Markov Decision Process (MDP), $(\mathcal{S}, \mathcal{A}, \mathcal{T}, r, \gamma)$, where $\mathcal{S}$ is the state space, $\mathcal{A}$ the action space, $\mathcal{T}$ the transition dynamics, $r$ the reward, and $\gamma$ the discount factor.

- **State:** At step $t$, the state is defined as the current query under resolution, denoted by $s_t = q_t$. This formulation emphasizes that the environment is always centered on the query being processed at step $t$, while contextual signals are implicitly captured through the evolving workflow structure.

- **Action:** Without loss of generality, we define the agent role set as {planner, executor, summarizer}, following prior multi-agent designs (Wu et al., 2024; Chen et al., 2023a; Barachini & Stary, 2022; Tran et al., 2025). Each action is a pair $a_t = (\alpha_t, m_t)$, where $\alpha_t$ specifies the role and $m_t$ indexes one of the $K$ candidate LLM backbones, yielding $|\mathcal{A}| = 3K$ possible actions. In brief, the *planner* decomposes a complex query into atomic sub-queries; the *executor* generates responses with or without contextual grounding; and the *summarizer* condenses multiple outputs into a coherent and concise answer. To ensure semantic validity, we impose a dynamic mask $M_t \subseteq \mathcal{A}$ restricting available actions:

  (1) At the first step, $M_0 = \{(\text{planner}, m), (\text{executor}, m) \mid m = 1, \ldots, K\}$, prohibiting summarizer choices.

  (2) At the final step, $M_T = \{(\text{executor}, m) \mid m = 1, \ldots, K\}$, enforcing workflow termination only by execution.

  (3) During the episode, planner actions are constrained by a hyperparameter $P_{\max} \in \mathbb{N}$ such that if $\sum_{i=0}^{t} \mathbf{1}(\alpha_i = \text{planner}) \geq P_{\max}$, all planner actions are removed from $M_{t+1}$.

  Thus, the effective policy $\pi : \mathcal{S} \to M_t$ always selects semantically valid actions.

- **Transition:** The transition dynamics update the workflow memory by determining both the next query to resolve and the observable response at step $t$. Formally, the environment outputs $(s_{t+1}, o_t) = \mathcal{T}(s_t, a_t)$, where $o_t$ denotes the response generated by action $a_t$ on the current query $s_t$. Concretely:

  (1) If $\alpha_t = \text{planner}$, the current query is decomposed into sub-queries, $o_t$ is the set of newly created sub-queries, and $s_{t+1}$ is set to the first child query.

  (2) If $\alpha_t = \text{executor}$, the current query is resolved, $o_t$ is the generated answer, and $s_{t+1}$ moves to the next pending query (or terminates if $t = T$).

  (3) If $\alpha_t = \text{summarizer}$, the system aggregates completed responses, $o_t$ is the generated summary, and $s_{t+1}$ is set to the summary query.

Thus, the state always denotes the query under resolution, while the sequence of responses $\{o_t\}$ provides the observable outputs that accumulate along the trajectory to form the final answer.

- **Reward:** The reward balances task utility and routing cost:

$$r_t = \begin{cases} U(\hat{y}, y^*) - \alpha\, C(a_t), & \text{if } t = T \text{ (terminal)}, \\ -\alpha\, C(a_t), & \text{if } t < T \text{ (intermediate)}, \end{cases} \tag{4}$$

where $\hat{y}$ is the predicted output, $y^*$ the ground-truth label, $U(\hat{y}, y^*)$ a task-specific utility (e.g., accuracy, BLEU, or MRR), $C(a_t)$ the computational cost of action $a_t$, and $\alpha > 0$ a cost–utility trade-off coefficient.

- **Episode and Objective:** An episode terminates once the root query is resolved, i.e., $s_T \in \mathcal{S}_{\text{terminal}}$ for some finite $T$. The router seeks a policy maximizing the expected discounted return:

$$\max_{\pi}\ \mathbb{E}_{q \sim \mathcal{Q}} \left[ \sum_{t=0}^{T} \gamma^t r(s_t, a_t) \right], \quad a_t \sim \pi(s_t), \tag{5}$$

where $\mathcal{Q}$ is the query distribution and $\gamma \in (0, 1]$ the discount factor.

## 3.2 Heterogeneous Graph-based Policy Network

We parameterize the policy $\pi(a_t \mid s_t)$ using a heterogeneous graph neural network, denoted as `GARNet`. At each step $t$, the environment is represented as the union of a workflow memory graph and a historical memory graph: $\mathcal{G}_t = \mathcal{G}_{workflow} \cup \mathcal{G}_{history}, \mathcal{G}_t = (\mathcal{V}_t, \mathcal{E}_t)$.

**Node initialization.** We distinguish two types of graphs. For the workflow memory graph $\mathcal{G}_{workflow}$, the nodes are: $x_q \in \mathbb{R}^{d_q}$, $x_r \in \mathbb{R}^{d_r}$, $x_m = [e_{\text{role}}; U; C] \in \mathbb{R}^{d_m}$, where $x_q$ is the Longformer embedding of the current query, $x_r$ is the embedding of the response, and $x_m$ is the role hub node, constructed by concatenating the LLM-role textual embedding with task utility $U$ and cost $C$. For the historical memory graph $\mathcal{G}_{history}$, the nodes are: $x_{hq} \in \mathbb{R}^{d_q}$, $x_{hr} \in \mathbb{R}^{d_r}$, $x_m \in \mathbb{R}^{d_m}$, where $x_{hq}$ and $x_{hr}$ are embeddings of past queries and responses, and $x_m$ is the same role hub node shared across workflow memory and historical memory, providing a bridge for information exchange between the two graphs. To make this design explicit, GraphPlanner shown in Figure 2 maintains a fixed set of role hub nodes, one for each (LLM, role) pair, which are reused across all routing steps and across both $\mathcal{G}_{workflow}$ and $\mathcal{G}_{history}$. Every query and response node, regardless of which round they are generated in, connects to these same role hub nodes. This architecture aggregates three sources of information through a single interface: (i) the current workflow, (ii) accumulated historical interaction signals, and (iii) role-specific utility–cost profiles. By serving as the structural and semantic anchor between the two graphs, the shared role hub nodes enable consistent message passing, facilitate multi-round reasoning, and prevent the creation of redundant role nodes at each routing step.

**Graph construction.** In $\mathcal{G}_{workflow}$, queries are connected to roles through edges $e_{q-m}$, enriched with task performance and cost information. Responses are linked to the roles that generate them, and query–response edges preserve semantic alignment. In $\mathcal{G}_{history}$, historical queries $x_{hq}$ and responses $x_{hr}$ are connected to role hub nodes $x_m$ through edges $e_{hq-m}$ and $e_{hr-m}$. These encode accumulated experience about how roles performed in past interactions, which can influence the current workflow. The shared role hub nodes $x_m$ act as the anchor between $\mathcal{G}_{workflow}$ and $\mathcal{G}_{history}$, ensuring that decision-making at the current step benefits from both local memory and historical memory. Multi-round routing does not introduce additional role nodes. Each newly generated sub-query or response in later rounds is simply appended to the workflow graph and connected to the same shared role hub nodes. This implicitly connects different rounds through shared neighbors rather than explicit temporal edges, enabling GARNet to reuse accumulated knowledge throughout multi-step routing.

**Message passing.** Each node embedding is projected into a hidden space: $h_v^{(0)} = W_{\tau(v)} x_v, \ v \in \mathcal{V}_t$, where $\tau(v)$ denotes the node type. Messages are aggregated from neighbors: $m_v = \text{AGG}\{h_u^{(0)} : u \in N(v)\}$, and node states are updated via a residual connection: $h_v = h_v^{(0)} + \beta \cdot m_v$.

Table 2: **Phase 1 Evaluation: Model performance comparison with router baselines across five scenarios.** Phase 1 focuses on optimizing agentic routing within existing LLM workflows. We report results under two settings: Depth = 1, Width = 3 (left) and Depth = 2, Width = 2 (right). **Bold** and underline indicate the best and second-best results. Note that (*) indicates each single-round router is applied to select the LLM backbone for every agent in the Phase-1 workflow.

(a) Depth=1, Width=3

| Router | Math | Code | CS | WK | Popular | Average | | |
|---|---|---|---|---|---|---|---|---|
| | Acc | Acc | Acc | Acc | Acc | Acc | Cost | ΔAcc (%) |
| Router-KNN* | 48.11% | 70.00% | 84.67% | 29.41% | 27.00% | 54.80% | 1508.88 | +8.20 |
| Router-MLP* | 39.62% | 58.00% | 80.67% | 18.00% | 24.00% | 47.40% | 463.82 | +0.80 |
| Router-SVM* | 29.25% | 57.00% | 80.67% | 27.91% | 22.00% | 46.60% | 577.65 | 0.00 |
| RouterDC* | 41.51% | 52.00% | 85.33% | 25.00% | 30.00% | 50.30% | 1689.25 | +3.70 |
| GraphRouter* | 41.51% | 48.00% | 59.33% | 29.53% | 44.14% | 45.80% | 797.35 | -0.80 |
| GraphPlanner | 55.00% | 72.00% | 76.62% | 33.00% | 47.00% | 58.60% | 900.40 | +12.00 |

(b) Depth=2, Width=2

| Router | Math | Code | CS | WK | Popular | Average | | |
|---|---|---|---|---|---|---|---|---|
| | Acc | Acc | Acc | Acc | Acc | Acc | Cost | ΔAcc (%) |
| Router-KNN* | 66.04% | 63.00% | 75.33% | 32.91% | 33.56% | 56.20% | 2719.96 | +7.49 |
| Router-MLP* | 42.45% | 53.00% | 80.67% | 24.94% | 27.00% | 48.73% | 813.80 | +0.02 |
| Router-SVM* | 52.34% | 49.74% | 66.00% | 32.24% | 34.34% | 48.71% | 844.90 | 0.00 |
| RouterDC* | 36.79% | 50.00% | 84.00% | 23.00% | 33.00% | 48.74% | 2987.11 | +0.03 |
| GraphRouter* | 37.74% | 57.33% | 79.63% | 37.05% | 34.46% | 49.20% | 1215.32 | +0.49 |
| GraphPlanner | 66.50% | 70.00% | 77.00% | 37.50% | 45.00% | 60.40% | 1500.27 | +11.69 |

Table 3: **Phase 2 Evaluation: Model performance comparison with router baselines across five scenarios.** Phase 2 focuses on generating optimal workflows by jointly determining agent selection and LLM backbones. **Bold** and underline indicate the best and second-best results.

| Setting | Math | | Code | | CS | | WK | | Popular | | Average | | | |
|---|---|---|---|---|---|---|---|---|---|---|---|---|---|---|
| | Acc | Cost | Acc | Cost | Acc | Cost | Acc | Cost | Acc | Cost | Acc | Cost | Avg. LLM Calls | ΔAcc (%) |
| **Single-round Router** | | | | | | | | | | | | | | |
| Router-KNN | 40.4% | 183.5 | 66.0% | 236.8 | 82.0% | 105.8 | 27.0% | 119.5 | 17.0% | 232.3 | 49.7% | 169.2 | 1 | +9.3 |
| Router-MLP | 43.3% | 183.4 | 67.0% | 240.6 | 82.0% | 103.9 | 25.0% | 120.7 | 7.0% | 225.5 | 48.2% | 168.4 | 1 | +7.8 |
| Router-SVM | 38.6% | 185.0 | 58.0% | 254.0 | 78.0% | 104.0 | 13.0% | 136.0 | 13.0% | 220.0 | 45.4% | 179.8 | 1 | +5.0 |
| RouterDC | 57.6% | 186.7 | 51.0% | 99.2 | 79.3% | 39.4 | 32.0% | 142.1 | 39.0% | 272.7 | 54.3% | 138.7 | 1 | +13.9 |
| GraphRouter | 53.2% | 203.0 | 59.0% | 280.0 | 82.7% | 97.0 | 28.0% | 60.0 | 21.0% | 252.0 | 51.9% | 178.4 | 1 | +11.5 |
| **Multi-round Router** | | | | | | | | | | | | | | |
| Prompt LLM | 37.7% | 1154.8 | 56.0% | 954.3 | 76.2% | 1215.6 | 24.0% | 798.1 | 10.0% | 1238.4 | 40.8% | 1070.4 | 12.5 | +0.4 |
| Router-KNN-MR | 39.6% | 407.2 | 53.0% | 432.6 | 73.5% | 266.4 | 24.0% | 327.9 | 12.0% | 303.7 | 40.4% | 347.6 | 7.2 | 0.0 |
| R2-Reasoner | 52.7% | 760.0 | 49.6% | 1200.0 | 72.8% | 380.0 | 27.1% | 270.0 | 37.4% | 740.0 | 50.1% | 643.6 | 5.4 | +9.8 |
| Router-R1 | 45.3% | 46.4 | 52.0% | 74.5 | 81.2% | 27.9 | 28.6% | 57.7 | 37.2% | 199.0 | 51.8% | 76.3 | 1.8 | +11.4 |
| GraphPlanner | 67.0% | 682.2 | 76.0% | 1130.9 | 78.0% | 361.7 | 38.0% | 252.8 | 52.0% | 719.3 | 63.6% | 605.0 | 8.1 | **+23.2** |

**Nested dual-graph encoding.** We employ a dual-graph encoding scheme. First, the historical memory graph is encoded: $H^{(\mathrm{his})} = \mathtt{GARNet}_{\theta^{\mathrm{his}}}(\mathcal{G}_{history})$, producing updated embeddings of the role hub nodes summarizing past query–response interactions. These are then injected into the workflow memory graph encoder: $H^{(\mathrm{loc})} = \mathtt{GARNet}_{\theta^{\mathrm{loc}}}(\mathcal{G}_{workflow}; H^{(\mathrm{his})})$, yielding local-contextualized representations of queries, roles, and responses.

This shared-node mechanism ensures that the role hub nodes encode both the accumulated historical interaction patterns and the evolving workflow state. Because every query and response, across all routing rounds, attaches to the same set of role hubs, GARNet can naturally integrate multi-round contextual information without requiring an explicit temporal graph structure.

**State fusion and action scoring.** The global state representation is obtained by fusing the current query embedding $s_t$, $z_t = f_{\mathrm{trans}}(s_t) \in \mathbb{R}^d$. Each candidate action corresponds to a LLM-role node embedding $h_{m,j} \in H^{(\mathrm{loc})}$. Compatibility scores are computed as $\mathrm{score}_j = z_t^{\top} h_{m,j}$, masked by $M_t$, and normalized into a probability distribution: $\pi(a_t = j \mid s_t) = \frac{\exp(\mathrm{score}_j)\cdot\mathbf{1}\{a_j \in M_t\}}{\sum_k \exp(\mathrm{score}_k)\cdot\mathbf{1}\{a_k \in M_t\}}$.

**GraphPlanner Training**. We optimize the heterogeneous graph-based policy network using Proximal Policy Optimization (PPO) (Schulman et al., 2017), a widely used actor–critic reinforcement learning algorithm. More details can be found in Appendix B.

## 4 EXPERIMENTS

In this section, we conduct a comprehensive evaluation of GraphPlanner across a wide range of tasks spanning multiple domains, comparing its performance against both single-round and multi-round routers. We begin by briefly outlining the experimental settings. More implementation details can be found in Appendix C.

**Dataset**. We evaluate router models on 14 tasks across 6 domains (including in-domain and out-of-domain evaluation), selected from recent influential reports on LLM evaluation (Anthropic, 2024; Yang et al., 2025; Gunter et al., 2024). Following prior work (Chen et al., 2023a; Feng et al., 2025), we curated training and test splits for each task (details in Appendix D). The in-domain evaluation datasets include: **(1) Math**: GSM8K (Cobbe et al., 2021) and MATH (Hendrycks et al., 2021b). **(2) Code**: MBPP (Austin et al., 2021) and HumanEval (Chen et al., 2021). **(3) Commonsense Reasoning**: CommonsenseQA (Talmor et al., 2019), ARC (Clark et al., 2018), and OpenBookQA

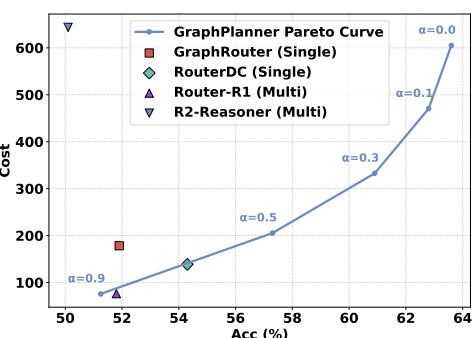

Figure 3: **Detailed illustration of Phase 1 Evaluation.** In the Phase 1 evaluation, we primarily assess how well `GraphPlanner` optimizes user-defined LLM workflows. Without loss of generality, we construct the graph-based LLM workflow shown above and set two hyperparameters: Depth and Width. Here, Depth refers to the number of planners, and Width denotes the maximum number of sub-queries that each planner is allowed to decompose.

(Mihaylov et al., 2018). **(4) World Knowledge**: NaturalQuestions (Kwiatkowski et al., 2019) and TriviaQA (Joshi et al., 2017). **(5) Popular**: MMLU (Hendrycks et al., 2021a) and GPQA (Rein et al., 2023). We further include **(6) Out-of-domain evaluation**, including LogicGrid (Mitra & Baral, 2015), MGSM (Shi et al., 2022), and CommonGen (Lin et al., 2019), which target reasoning, multilingual generalization, and commonsense generation, and are used only for evaluation, ensuring the router is tested on genuinely unseen domains to rigorously assess generalization.

**LLM backbone**. Following previous work (Feng et al., 2025), we employed 12 representative LLMs grouped into three scales: *small*, *medium*, and *large*, including **(1) Small scale LLMs:** Qwen2.5 (7b) (Qwen et al., 2025), CodeGemma (7b) (Team et al., 2024a), Mistral (7b) (Jiang et al., 2023), LLaMA-3.1 (8b) (Grattafiori et al., 2024), LLaMA-3 ChatQA (8b) (Liu et al., 2024), and Gemma-2 (9b) (Team et al., 2024b); **(2) Medium scale LLMs:** LLaMA-3.3 Nemotron Super (49b) (Wang et al., 2024b), LLaMA-3.1 Nemotron (51b) (Wang et al., 2024b), and LLaMA-3 ChatQA (70b) (Liu et al., 2024); **(3) Large scale LLMs:** Mixtral (8×22b) (Jiang et al., 2024). We further summarize the corresponding scales, input price, and output price of each LLM in Table 10 in the Appendix. Notably, besides the above LLMs that are involved in training, three models—*Mistral-Nemo (12b)* (Mistral AI, 2024), *Mixtral (8×7b)* (Jiang et al., 2024), and *Mixtral (8×22b)* (Jiang et al., 2024)—are deliberately withheld. These underlined models are reserved exclusively for evaluation, ensuring that the assessment rigorously reflects the router's generalization ability to previously unseen LLMs across different scales. More details can be found in Table 7 in the Appendix.

**Task description**. We designed a two-phase evaluation. As shown in Figure 3, **Phase 1 Evaluation** focuses on optimizing agentic routing within the user-predefined LLM workflows. In this phase, we specify different widths and depths for agentic workflows. The task is: given a query, different routers are expected to optimize the choice of LLM backbones for different agents. In particular, we conduct experiments mainly under two settings: Depth = 1, Width = 3 and Depth = 2, Width = 2. Here, depth refers to the number of planners, and width denotes the maximum number of sub-queries that each planner is allowed to decompose. **Phase 2 Evaluation** focuses on generating optimal workflows. Here, given a query, different routers are expected to simultaneously optimize both the agent selections and the corresponding LLM backbones. **Baselines and metrics.** We evaluate a variety of baseline methods across 6 scenarios. The baselines are categorized into two groups: **(a)** *Single-round routers* that route a query by calling an LLM once, and **(b)** *Multi-round routers* that solve a query by calling multiple LLMs. For all routers, following previous work (Feng et al., 2025), we use *Acc* and *Cost* to evaluate routing performance. Here, *Acc* refers to the task-specific evaluation metric introduced in Table 9 of the Appendix. *Cost* is calculated with the number of input tokens

Figure 4: **Compared to baseline routers, `GraphPlanner` consistently forms the Pareto frontier, offering more efficient trade-offs between Acc and Cost.** `GraphPlanner` (with $\alpha \in \{0.0, 0.1, 0.3, 0.5, 0.9\}$) is compared against two single-round routers and two multiple-round routers.

Table 4: **Comparison of tokens used, GPU compute, and average LLM calls in Phase-2 training.** We observe that, compared with other routers, GraphPlanner not only reduces token consumption during training but also lowers GPU compute requirements.

| Router | Used Tokens | GPU Compute | Avg. LLM Train Calls |
|---|---|---|---|
| GraphRouter | 64.87M | 1.54GiB | 1 |
| RouterDC | 64.87M | 10.56GiB | 1 |
| Router-R1 | 150.36k | 186.26GiB | 1.18 |
| GraphPlanner | 182.45k | 1.04GiB | 4.25 |

Table 5: **Performance on unseen datasets LogicGrid, MGSM, and CommonGen in Phase-2.** We report both the individual results on each dataset and the averaged performance across them to evaluate the router's zero-shot generalization ability on unseen datasets.

| Router | LogicGrid | MGSM | CommonGen | Avg. Acc |
|---|---|---|---|---|
| GraphRouter | 12% | 68% | 57% | 46% |
| RouterDC | 32% | 82% | 60% | 58% |
| Router-R1 | 24% | 40% | 48% | 38% |
| GraphPlanner | 60% | 92% | 82% | 78% |

and output tokens and the cost of different LLMs in Table 10 of the Appendix. Here we utilize GPT-2 as in (Feng et al., 2024) to calculate the number of tokens. Specifically, we have: **(a) Single-round routers.** We consider five representative single-round routers: 1) *RouterKNN* (Shnitzer et al., 2023), a non-parametric baseline that assigns a query to the nearest neighbors in embedding space and predicts the majority LLM label; 2) *RouterMLP* (Shnitzer et al., 2023), a multi-layer perceptron that leverages query embeddings and task context for routing; 3) *RouterSVM* (Hu et al., 2024a), a support vector machine trained on query features and task labels; 4) *RouterDC* (Chen et al., 2024a), a query-based router trained with dual contrastive learning over encoder and LLM embeddings, designed to distinguish among multiple LLMs even when several perform well; 5) *GraphRouter* (Feng et al., 2024), a graph-based model that formulates routing as node classification over a heterogeneous graph of queries, tasks, and LLMs with learned edge interactions. **(b) Multi-round routers.** We consider four representative multi-round routers: 1) *Prompt LLM* (Zhang et al., 2025), a baseline that directly prompts an LLM to select LLMs without explicit routing modules, serving as a simple multi-round strategy; 2) *Router-KNN-MR* (Zhang et al., 2025), an iterative extension of Router-KNN that repeatedly queries nearest neighbors in embedding space to refine routing decisions; 3) *R2-Reasoner* (Shao et al., 2025), a reasoning-oriented router that conducts multi-step internal deliberation before invoking experts, improving decision quality through structured reasoning; 4) *Router-R1* (Zhang et al., 2025), the proposed reinforcement learning framework that interleaves think and route actions, aggregates expert outputs across rounds, and optimizes routing with a reward function balancing accuracy and cost.

## 4.1 GRAPHPLANNER OUTPERFORMS SINGLE-ROUND AND MULTI-ROUND ROUTERS

For each setting in Phase 1 and Phase 2, we train and test a unified GraphPlanner across all scenarios. We compare GraphPlanner with five single-round routers and four multi-round routers, and report the results of Phase 1 and Phase 2 in Table 2 and Table 3, respectively. We have the following observations.

Specifically, for Phase-1, since there are no existing baselines, we extend the aforementioned single-round routers to the Phase-1 setting for comparison. Specifically, we first train these single-round routers on the dataset reported in Table 6. During inference, each router is applied to select the LLM backbone for every agent within the Phase-1 workflow. To distinguish them from their original usage, we append an asterisk (*) to the single-round routers when they are adapted to the Phase-1 setting.

**GraphPlanner attains SOTA results across diverse scenarios**. Across both phases, GraphPlanner demonstrates clear superiority over competitive baselines. In Phase 1, it achieves SOTA results in four out of five tasks while maintaining the highest overall average accuracy, yielding a minimum improvement of +3.8% compared to the strongest baseline. In Phase 2, GraphPlanner again secures SOTA in four out of five tasks and remains highly competitive in the remaining one, with an overall accuracy gain of +9.3% over the best baseline. These findings underscore GraphPlanner's robustness and effectiveness across diverse routing scenarios. We can also observe that Phase 2 further amplifies GraphPlanner's advantage: its average accuracy surpasses the best Phase-1 results by about 5%, showing that the ability to construct query-specific optimal agentic workflows yields stronger performance than optimizing within fixed workflows. The improvements are especially pronounced in reasoning-oriented tasks such as Math and Code, with gains of 5.0% and 4.0%, because these domains demand multi-step planning and benefit substantially from adaptive agent structures. By contrast, recognition-focused tasks show only modest increases of around 1.0%, since they rely more on straightforward pattern matching, where flexible workflow exploration provides limited additional benefit.

**GraphPlanner achieves superior routing performance with reduced training compute and lower token cost**. We further analyzed the training overhead of GraphPlanner compared with

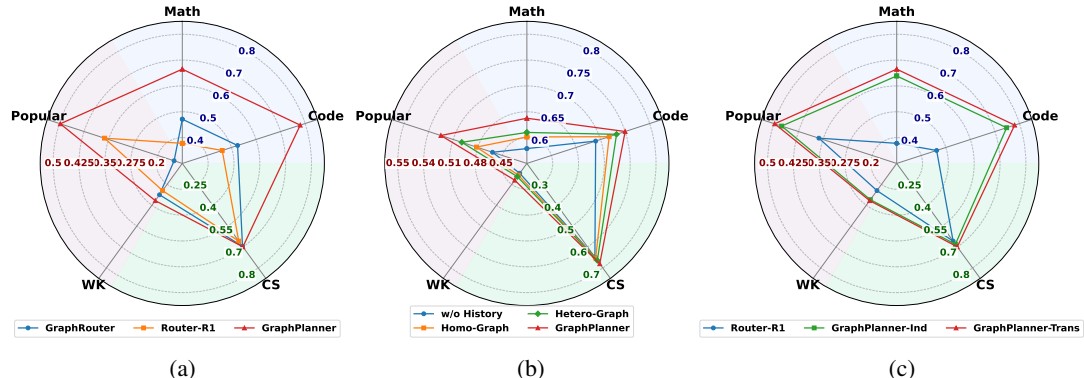

(a)    (b)    (c)

Figure 5: **Comparison of `GraphPlanner` against baselines across different experimental settings in five scenarios under Phase-2.** (a) *Unseen LLMs generalization:* We add the unseen LLMs—not introduced in the training in Table 7—into the LLM pool, and then evaluate the zero-shot generalization ability of `GraphPlanner`, compared with GraphRouter and Router-R1. (b) *Historical memories utilization ablations:* We ablate historical memories utilization, contrasting `GraphPlanner` with variants w/o History, Homo-Graph, and Hetero-Graph encodings. (c) *Transductive vs. Inductive routing inference:* We analyze `GraphPlanner` under transductive vs. inductive settings, where `GraphPlanner` consistently outperforms best multi-round router Router-R1.

other routers. In the Phase-2 training, we compared `GraphPlanner` with several representative routers in terms of tokens used, GPU compute, and average LLM calls, as shown in Table 4. We observe that `GraphPlanner` achieves the smallest GPU compute among all routers, demonstrating the efficiency of its lightweight design. Moreover, although `GraphPlanner` consumes slightly more tokens than Router-R1, the results on average LLM training calls indicate that this is due to `GraphPlanner` performing more extensive multi-step planning for different queries during training, which in turn leads to better routing performance.

**`GraphPlanner` effectively balances trade-off between performance and cost**. As shown in Figure 4, `GraphPlanner` consistently forms the Pareto frontier, surpassing both single-round and multi-round routers. By adjusting $\alpha$, it flexibly shifts between high-Acc, high-Cost, and low-Cost, lightweight settings. Compared with baselines, `GraphPlanner` achieves either higher Acc under the same Cost or lower Cost at the same Acc, demonstrating more efficient and controllable trade-offs.

### 4.2 GRAPHPLANNER NICELY GENERALIZES ACROSS UNSEEN TASKS AND LLMS

A key challenge for router design is whether the learned strategy can generalize beyond the training distribution, adapting to entirely new tasks or unseen LLM backbones. To this end, we evaluate `GraphPlanner` in a zero-shot setting on both novel tasks and unseen LLMs, analyzing its robustness and adaptability under Phase-2.

**`GraphPlanner` generalizes robustly to unseen tasks**. As shown in Table 5, `GraphPlanner` demonstrates strong zero-shot generalization, achieving an average Acc of 78% across LogicGrid, MGSM, and CommonGen. This significantly outperforms both single-round routers (GraphRouter 46%, RouterDC 58%) and the multi-round router Router-R1 (38%). Notably, `GraphPlanner` achieves the highest performance on each dataset (60% on LogicGrid, 92% on MGSM, and 82% on CommonGen), underscoring its robustness in handling diverse unseen tasks without additional tuning.

**`GraphPlanner` effectively adapts to unseen LLMs in a zero-shot setting**. As illustrated in Figure 5(a), `GraphPlanner` demonstrates strong adaptability when evaluated with unseen LLMs not introduced during training. Compared with GraphRouter and Router-R1, `GraphPlanner` consistently achieves superior performance across all task domains, indicating that its routing strategy generalizes effectively to new backbone models without additional fine-tuning. This highlights the robustness of `GraphPlanner` in handling zero-shot scenarios where the underlying LLMs differ from those seen in training.

### 4.3 ABLATION STUDIES VALIDATE GRAPHPLANNER'S KEY COMPONENTS

To better understand the contributions of individual design choices within `GraphPlanner`, we conduct ablation studies by systematically removing or modifying key components. These experiments allow us to isolate the impact of historical interaction modeling and different routing inference strategies, thereby validating the necessity and effectiveness of each module.

**GARNet leverages historical agentic LLM interactions and current agent workflow states to enhance `GraphPlanner`'s decision-making**. To assess the role of historical memories in `GraphPlanner`, we design three ablation variants:

- **w/o History**: Removes all historical states, forcing `GraphPlanner` to rely solely on the current input without accumulated interaction context.
- **Homo-Graph**: Replaces `GARNet` with a homogeneous graph neural network[1] that treats all nodes and edges are treated as the same type, capturing structural relations but discarding role-specific heterogeneity.
- **Hetero-Graph**: Replaces `GARNet` with a heterogeneous graph neural network[2] where nodes and edges are assigned different types, which distinguishes among roles but does not incorporate workflow dynamics.

As shown in Figure 5(b), removing historical memories (*w/o History*) leads to a substantial performance drop, demonstrating that accumulated interactions provide indispensable contextual signals beyond single-step reasoning. Introducing graph structures partially mitigates this degradation: the Homo-Graph variant captures basic relational structure but lacks role differentiation, yielding only limited gains. The Hetero-Graph variant consistently outperforms Homo-Graph by distinguishing among agent roles, confirming that heterogeneity carries richer relational cues. Nevertheless, both graph-based variants remain clearly inferior to the full `GARNet` design. Beyond heterogeneous modeling, `GARNet` provides an efficient and lightweight mechanism to capture workflow dynamics, enabling it to model not only who interacts but also how these interactions evolve over time. This dynamic perspective equips `GraphPlanner` with stronger contextual awareness and adaptability, allowing it to leverage historical interactions far more effectively than generic GNN-based encoders.

**`GraphPlanner` generates routing decisions under both inductive and transductive ways**. To evaluate the effect of different routing inference strategies, we compare two settings:

- **Inductive**: During inference, `GraphPlanner` directly generates routing decisions without holding out or reusing any historical interactions from the training phase. This design is lightweight and avoids additional storage or retrieval overhead.
- **Transductive**: During inference, `GraphPlanner` leverages preserved historical interactions collected during training, enabling richer context utilization at the cost of higher computational and memory overhead.

As shown in Figure 5(c), the transductive strategy achieves slightly better overall performance, demonstrating that leveraging stored historical interactions provides additional contextual cues that enhance routing quality. However, this improvement comes with increased inference cost, as the model must maintain and query interaction histories. The inductive strategy, while more lightweight, still maintains strong performance and consistently outperforms the best multi-round router baseline, Router-R1. In summary, both inference strategies are valuable: the transductive setting delivers the highest accuracy when efficiency is less critical, whereas the inductive setting provides a more resource-efficient solution with competitive performance. This flexibility allows `GraphPlanner` to adapt to different user priorities, offering either maximum effectiveness or efficient deployment without significant performance sacrifice.

## 5 CONCLUSION

We introduced `GraphPlanner`, a heterogeneous graph-based multi-agent router that casts routing as workflow generation within an MDP, leveraging the heterogeneous graph `GARNet` to integrate historical and workflow memories and training the policy via reinforcement learning. Extensive experiments across 14 tasks and 6 domains show that `GraphPlanner` delivers state-of-the-art performance, robust generalization to unseen tasks and LLMs, and favorable trade-offs between accuracy and computational cost. These results underscore the potential of extending LLM routing into agentic settings and open new directions for scalable, cooperative multi-agent LLM systems. In future work, we plan to incorporate richer agent profiles beyond Planner, Executor, and Summarizer to further enhance agentic routing.

---

[1] https://pytorch-geometric.readthedocs.io/en/latest/generated/torch_geometric.nn.conv.GCNConv.html

[2] https://pytorch-geometric.readthedocs.io/en/2.5.3/generated/torch_geometric.nn.conv.HeteroConv.html

ETHICS STATEMENT

All authors of this paper have read and adhered to the ICLR Code of Ethics. Our work does not involve human subjects, personal data, or sensitive attributes. We followed best practices for data usage, ensured compliance with licensing terms, and considered potential risks of bias or misuse.

REPRODUCIBILITY STATEMENT

We have made every effort to ensure the reproducibility of our results. Details of the model architecture, training settings, and hyperparameters are described in Section 4. All datasets we used are publicly available. The training scripts and evaluation code will be released upon publication to facilitate replication.

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

## A   ADDITIONAL RELATED WORK

**LLM-Agents and Agentic Systems**. Recent studies have shown that organizing LLM-based agents into multi-agent systems (MAS) can substantially enhance reasoning, adaptability, and overall performance beyond single-agent settings (Wang et al., 2024a; Qian et al., 2024; Guo et al., 2024). Early frameworks such as AutoGen (Wu et al., 2024), LLM-Debate (Du et al., 2023), and AgentVerse (Chen et al., 2023a) demonstrated gains in factuality, robustness, and efficiency, but relied on manually designed protocols that limited adaptability (Zhuge et al., 2024; De Zarzà et al., 2023). Moreover, most MAS assume agents share the same backbone, constraining heterogeneity where diverse models could provide complementary strengths. Inspired by human teamwork, later work explored autonomous cooperation, showing that agents can self-organize, exhibit emergent behaviors, and dynamically divide labor (Barachini & Stary, 2022; Tran et al., 2025). Studies further reported improved reasoning through social behaviors, negotiation, and role specialization (Zhang et al., 2023; Chen et al., 2024b; Chang, 2025). These advances highlight a shift toward automated agentic systems, yet current MAS research predominantly relies on identical LLM backbones across all agents, which fundamentally constrains the exploration of agent capability diversity and limits the potential for truly complementary collaboration. To address these limitations, a newer line of work focuses on automating the design of multi-agent workflows rather than relying on rigid, manually constructed protocols. Recent workflow-generation systems such as ADAS (Hu et al., 2024b) adaptively schedule agents based on task complexity, AFlow (Zhang et al., 2024a) automatically searches action graphs to construct multi-step workflows, and AgentSquare (Shang et al., 2024) generates task-specific collaboration strategies from a library of roles, but all still assume largely homogeneous agent capabilities. Inspired by the potential of recent workflow-generation systems to better structure multi-step reasoning, we introduce an agentic routing framework that automatically selects heterogeneous LLM agents and composes workflows tailored to each query.

**Tool-Augmented LLMs and Real-World Agent Ecosystems**. A parallel line of work focuses on deploying LLM agents in real-world tool ecosystems, where models interact with external APIs, knowledge bases, or software environments. Early systems such as Toolformer (Schick et al., 2023), Gorilla (Patil et al., 2024), and ReAct-based tool agents (Yao et al., 2022) enable LLMs to invoke functions or APIs for grounded decision-making. Production frameworks—including LangChain, Semantic Kernel, AutoGPT, and OpenAI's function-calling agents—further demonstrate the importance of reliable tool integration, execution monitoring, and safety in practical deployments. These systems highlight that tool-use and API grounding are central to building real-world agentic pipelines, yet they typically rely on static or manually designed workflows, underscoring the need for more flexible, learned workflow generation and routing strategies.

**LLM routers**. Routing among multiple LLMs is a key paradigm for balancing efficiency and accuracy. Existing approaches fall into single-round and multi-round routers. Single-round routers make one-shot assignments using query embeddings or classifiers, such as RouterKNN (Shnitzer et al., 2023) and RouterMLP (Shnitzer et al., 2023), RouterSVM (Hu et al., 2024a), RouterDC (Chen et al., 2024a), and GraphRouter (Feng et al., 2024). These methods are efficient but lack sequential reasoning. Multi-round routers enable iterative decisions, as in Prompt LLM (Zhang et al., 2025), Router-KNN-MR (Zhang et al., 2025), R2-Reasoner (Shao et al., 2025), and Router-R1 (Zhang et al., 2025), which combine deliberation and routing with reinforcement learning. While more flexible, they remain restricted to backbone selection without modeling agent roles or heterogeneity. Current paradigms thus face two limitations: focusing only on backbone choice and assuming homogeneous models. Agentic routing researched by our paper addresses these by jointly deciding which agent and which backbone to invoke, combining routing efficiency with the adaptability, specialization, and heterogeneity of multi-agent systems.

## B   GRAPHPLANNER TRAINING DETAILS

We optimize the heterogeneous graph-based policy network using Proximal Policy Optimization (PPO) (Schulman et al., 2017), a widely used actor–critic reinforcement learning algorithm. PPO trains the policy by maximizing:

$$\mathcal{L}^{\text{PPO}}(\theta) = \hat{\mathbb{E}}_t\Big[ \min\big(\rho_t(\theta)\hat{A}_t, \ \text{clip}(\rho_t(\theta), 1-\epsilon, 1+\epsilon)\hat{A}_t\big)\Big], \tag{6}$$

where $\pi_\theta$ and $\pi_{\theta^{\text{old}}}$ denote the current and previous policies, respectively, and

$$\rho_t(\theta) = \frac{\pi_\theta(a_t \mid s_t, \mathcal{G}_{\text{workflow}}, \mathcal{G}_{\text{history}})}{\pi_{\theta^{\text{old}}}(a_t \mid s_t, \mathcal{G}_{\text{workflow}}, \mathcal{G}_{\text{history}})}. \tag{7}$$

Here, $\hat{A}_t$ is the estimated advantage at step $t$, $\epsilon$ is a clipping threshold, $s_t$ the current state, $a_t$ the chosen action, $\mathcal{G}_{\text{workflow}}$ is the workflow interaction graph, and $\mathcal{G}_{\text{history}}$ is the historical interaction graph.

## C  IMPLEMENTATION DETAILS

We implement `GraphPlanner` with a PPO backbone, where both policy and value functions are parameterized by `GARNet` to integrate local and historical state information. Local state graphs encode query embeddings, role–LLM embeddings, and memory updates, while historical graphs aggregate past interaction representations; each graph is projected via a linear–normalization–ReLU block and fused by meta-key aggregation. `GARNet` is implemented using the `torch_scatter` library for efficient graph-based message passing and sparse aggregation. The policy network computes action probabilities by matching fused state representations (query, task, and state tower outputs) against role–LLM embeddings with action masking, while the value network processes state, local, and historical features through multi-layer transformations to output scalar value estimates. Training follows PPO with clipped objectives ($\gamma = 0.99$, $\epsilon = 0.2$, $k = 4$ epochs per update). We set hidden dimension to 32, candidate embedding dimension to 1536, and state embedding dimension to 768. Adam optimizer is used with learning rate $3 \times 10^{-4}$ for policy and doubled for value, combined with gradient clipping (norm 0.5), BF16 training, and gradient checkpointing. To improve data collection efficiency, we adopt a multi-threaded rollout design that processes multiple queries in parallel and generates routing interactions simultaneously. This design increases sample throughput, reduces wall-clock training time, and stabilizes PPO updates by providing more diverse experience per iteration. Training is capped at 1000 episodes with early stopping once policy entropy drops below a threshold, indicating reduced exploration. During evaluation, greedy decoding is applied and the best model is selected by running reward. All experiments are conducted on a single NVIDIA A6000 GPU.

## D  DATASET AND LLM BACKBONE DETAILS

Table 6: **The domains and corresponding tasks of the dataset used in our experiment.** Specifically, it spans 6 representative domains and 14 tasks. Note that the scenarios and corresponding tasks marked with underline are held out from the training set and reserved solely for evaluating the router's generalization performance on unseen tasks.

| Domain | Tasks |
|---|---|
| Math | GSM8K, MATH |
| Code | MBPP, HumanEval |
| Commonsense Reasoning | CommonsenseQA, ARC, OpenBookQA |
| World Knowledge | NaturalQuestions, TriviaQA |
| Popular | MMLU, GPQA |
| Out-of-domain Testing | LogicGrid, MGSM, CommonGen |

Table 7: **The scales and corresponding LLMs used in our experiment.** Specifically, the 12 LLMs are categorized into three scales based on model size. Note that the LLMs marked with underline are not involved in the training process, but are only included in experiments that evaluate the router's generalization to unseen LLMs.

| Scale | LLMs |
|---|---|
| Small | Qwen2.5 (7b), CodeGemma (7b), Mistral (7b) |
| | LLaMA-3.1 (8b), LLaMA-3 ChatQA (8b), Gemma-2 (9b) |
| | Mistral-Nemo (12b) |
| Medium | LLaMA-3.3 Nemotron Super (49b) |
| | LLaMA-3.1 Nemotron (51b), Mixtral (8x7b) |
| | LLaMA-3 ChatQA (70b) |
| Large | Mixtral (8x22b) |

### D.1  TASK DESCRIPTIONS

The benchmarks summarized in Tables 8 and 9 span math, code, commonsense reasoning, world knowledge, popular comprehensive tests, and out-of-domain evaluation. Below we provide brief descriptions for each task to orient the reader.

Table 8: **Sample counts in the training set and test set across different tasks.**

| Domain | Tasks | Train Cases | Test Cases |
|---|---|---|---|
| Math | GSM8K (Cobbe et al., 2021) | 500 | 50 |
| | MATH (Hendrycks et al., 2021b) | 500 | 50 |
| Code | MBPP (Austin et al., 2021) | 374 | 50 |
| | HumanEval (Chen et al., 2021) | 120 | 44 |
| Commonsense Reasoning | CommonsenseQA (Talmor et al., 2019) | 500 | 50 |
| | ARC (Clark et al., 2018) | 500 | 50 |
| | OpenBookQA (Mihaylov et al., 2018) | 500 | 50 |
| World Knowledge | NaturalQuestions (Kwiatkowski et al., 2019) | 500 | 50 |
| | TriviaQA (Joshi et al., 2017) | 500 | 50 |
| Popular | MMLU (Hendrycks et al., 2021a) | 500 | 50 |
| | GPQA (Rein et al., 2023) | 400 | 44 |
| Out-of-domain Testing | LogicGrid (Mitra & Baral, 2015) | 0 | 50 |
| | MGSM (Shi et al., 2022) | 0 | 50 |
| | CommonGen (Lin et al., 2019) | 0 | 50 |

Table 9: **The tasks and corresponding evaluation metrics of the dataset used in our experiment, organized by domain.**

| Domain | Tasks | Metrics |
|---|---|---|
| Math | GSM8K (Cobbe et al., 2021) | Accuracy |
| | MATH (Hendrycks et al., 2021b) | Accuracy |
| Code | MBPP (Austin et al., 2021) | Pass@1 |
| | HumanEval (Chen et al., 2021) | Pass@1 |
| Commonsense Reasoning | CommonsenseQA (Talmor et al., 2019) | Accuracy |
| | ARC (Clark et al., 2018) | Accuracy |
| | OpenBookQA (Mihaylov et al., 2018) | Accuracy |
| World Knowledge | NaturalQuestions (Kwiatkowski et al., 2019) | CEM |
| | TriviaQA (Joshi et al., 2017) | CEM |
| Popular | MMLU (Hendrycks et al., 2021a) | Accuracy |
| | GPQA (Rein et al., 2023) | Accuracy |
| Out-of-domain Testing | LogicGrid (Mitra & Baral, 2015) | Accuracy |
| | MGSM (Shi et al., 2022) | Accuracy |
| | CommonGen (Lin et al., 2019) | Coverage |

**GSM8K**. GSM8K is a grade-school math word-problem dataset designed to probe multi-step arithmetic reasoning with natural-language solutions (Cobbe et al., 2021). Problems typically require decomposing the question into several simple operations and tracking intermediate quantities. It has become a standard testbed for chain-of-thought prompting and verifier-based solution selection. We report accuracy following the setup in Table 9.

**MATH**. The MATH benchmark consists of 12,500 competition-style problems spanning algebra, geometry, number theory, and more, each with step-by-step solutions (Hendrycks et al., 2021b). It evaluates symbolic reasoning and solution derivation beyond simple calculation. Because problems include full worked solutions, the dataset also supports training methods that supervise intermediate reasoning. We report accuracy as in Table 9.

Table 10: **Language Models and estimated price (in $ per 1M tokens).**

| Size Type | Model | Size | Input Price | Output Price |
|---|---|---|---|---|
| Small | Qwen2.5 (Qwen et al., 2025) | 7B | 0.20 | 0.20 |
| | CodeGemma (Team et al., 2024a) | 7B | 0.20 | 0.20 |
| | Mistral (Jiang et al., 2023) | 7B | 0.20 | 0.20 |
| | LLaMA-3.1 (Grattafiori et al., 2024) | 8B | 0.20 | 0.20 |
| | LLaMA-3 ChatQA (Liu et al., 2024) | 8B | 0.20 | 0.20 |
| | Gemma-2 (Team et al., 2024b) | 9B | 0.20 | 0.20 |
| | Mistral-Nemo (Mistral AI, 2024) | 12B | 0.30 | 0.30 |
| Medium | LLaMA-3.3 Nemotron Super (Wang et al., 2024b) | 49B | 0.90 | 0.90 |
| | LLaMA-3.1 Nemotron (Wang et al., 2024b) | 51B | 0.90 | 0.90 |
| | Mixtral (Jiang et al., 2024) | 56B (8×7B) | 0.60 | 0.60 |
| | LLaMA-3 ChatQA (Liu et al., 2024) | 70B | 0.90 | 0.90 |
| Large | Mixtral (Jiang et al., 2024) | 176B (8×22B) | 1.20 | 1.20 |

**MBPP**. MBPP (Mostly Basic Python Problems) evaluates function-level code synthesis from short natural-language prompts (Austin et al., 2021). Tasks are designed to be solvable by entry-level programmers and include unit tests to automatically check correctness. It emphasizes core Python fluency, standard library use, and simple algorithmic reasoning. We use pass@1 as the principal metric (Table 9).

**HumanEval**. HumanEval measures functional correctness of generated Python code on hand-written problems with hidden unit tests (Chen et al., 2021). Prompts include function signatures and docstrings, and success requires passing all tests for a task. The benchmark introduced the widely used pass@k metric; we report pass@1 in Table 9. It stresses precise adherence to specifications and robust program synthesis.

**CommonsenseQA**. CommonsenseQA is a multiple-choice benchmark targeting commonsense reasoning via questions constructed from ConceptNet relations (Talmor et al., 2019). Distractors are chosen to be plausible, making surface cues insufficient. Models must draw on background knowledge and everyday plausibility. We report accuracy as listed in Table 9.

**ARC**. The AI2 Reasoning Challenge (ARC) comprises grade-school science questions split into Easy and Challenge subsets (Clark et al., 2018). The Challenge set contains items that defeat simple retrieval and co-occurrence methods, emphasizing multi-hop reasoning and science knowledge. Questions are multiple choice and text-only. Accuracy is reported per Table 9.

**OpenBookQA**. OpenBookQA evaluates the ability to apply a small "open book" of elementary science facts to novel situations (Mihaylov et al., 2018). Solving a question typically requires combining a core fact with commonsense or auxiliary knowledge. The format is multiple choice, and retrieval-augmented methods are commonly explored. We report accuracy as in Table 9.

**NaturalQuestions (NQ)**. NQ contains real, anonymized user queries paired with Wikipedia pages and annotated short and long answers (Kwiatkowski et al., 2019). It is a challenging, realistic QA benchmark requiring document-level comprehension and answer span identification. In our setup we evaluate case-insensitive exact match (CEM) following Table 9. The task stresses open-domain reading comprehension.

**TriviaQA**. TriviaQA provides questions written by trivia enthusiasts along with evidence documents, encouraging multi-sentence reasoning and robust retrieval (Joshi et al., 2017). Compared to earlier reading-comprehension datasets, it features more compositional and diverse questions. We report CEM as in Table 9. The dataset probes broad world knowledge under noisy evidence.

**MMLU**. MMLU (Massive Multitask Language Understanding) is a 57-subject multiple-choice exam spanning humanities, social sciences, STEM, and professional domains (Hendrycks et al., 2021a). It evaluates breadth of knowledge and reasoning in a zero- or few-shot setting. The benchmark is widely used for holistic comparison across models. We report accuracy per Table 9.

**GPQA**. GPQA (Graduate-Level Google-Proof Q&A) consists of expert-authored multiple-choice questions in biology, physics, and chemistry designed to resist simple web search (Rein et al., 2023).

It targets deep, specialized scientific understanding and careful reasoning. The dataset is intentionally difficult for both non-experts and strong LMs. We report accuracy as summarized in Table 9.

**LogicGrid**. This benchmark comprises classic logic-grid (Zebra-style) puzzles expressed in natural language, requiring deduction over entities, attributes, and constraints (Mitra & Baral, 2015). Success demands translating textual clues into structured constraints and performing consistent reasoning. It stresses symbolic consistency and global constraint satisfaction. We evaluate accuracy as in Table 9.

**MGSM**. MGSM (Multilingual Grade School Math) is a multilingual extension of GSM8K created by translating problems into diverse languages (Shi et al., 2022). It measures whether multi-step arithmetic reasoning ability transfers across scripts and linguistic structures. The benchmark is commonly used to assess chain-of-thought prompting in multilingual settings. We report accuracy per Table 9.

**CommonGen**. CommonGen evaluates generative commonsense reasoning by asking models to compose a coherent sentence that must include a given set of concepts (Lin et al., 2019). The task requires relational and compositional generalization beyond simple lexical co-occurrence. It is used to study controllable generation under semantic constraints. We report coverage per Table 9.

**AIME**. The American Invitational Mathematics Examination (AIME)[3] is a high-difficulty mathematical reasoning benchmark consisting of 15 integer-answer problems per year, designed to assess advanced problem-solving skills among top performers in the AMC competitions. Unlike typical multiple-choice math datasets, AIME items require multi-step symbolic reasoning, numeric precision, and long-horizon planning without external tools. The dataset evaluates models' abilities in algebra, geometry, number theory, and combinatorics, emphasizing deliberate reasoning over surface-level pattern matching. Following prior work, we adopt accuracy as the primary metric and report performance per Table 9.

## D.2 MODEL DESCRIPTIONS

The language models in Table 10 cover small, medium, and large configurations, with prices and sizes reported there. Below are brief descriptions to contextualize each model family.

**Qwen2.5 (7B)**. Qwen2.5 is a recent generation of the Qwen family, offering open-weight models optimized for general-purpose utility, instruction following, and strong reasoning/coding performance (Qwen et al., 2025). The 7B variant targets efficient deployment while retaining competitive capability across standard benchmarks. The family emphasizes multilingual coverage and long-context usability. We use the size and pricing shown in Table 10.

**CodeGemma (7B)**. CodeGemma is a code-specialized family derived from Gemma that supports code completion, generation, and conversational coding assistance (Team et al., 2024a). It adds training signals for software tasks and is commonly used with "fill-in-the-middle" prompting. The 7B model balances latency with solid pass@k performance on Python-centric benchmarks. Pricing details are given in Table 10.

**Mistral (7B)**. Mistral 7B is an open-weight, decoder-only transformer engineered for efficiency, featuring grouped-query attention and sliding-window attention for fast inference on long sequences (Jiang et al., 2023). Despite its compact size, it performs strongly on reasoning, math, and code tasks relative to larger predecessors. It is frequently used as a base for instruct-tuned and domain-specialized variants. See Table 10 for cost information.

**LLaMA-3.1 (8B)**. LLaMA-3.1 denotes Meta's open-weight models emphasizing improved instruction-following, multilinguality, and extended context capabilities (Grattafiori et al., 2024). The 8B model provides a lightweight option suitable for on-prem or edge use while retaining strong general performance. It is widely used as a base for fine-tuning and tool-using assistants. Pricing is listed in Table 10.

**LLaMA-3 ChatQA (8B / 70B)**. ChatQA refers to instruction-tuned QA/chat variants designed to excel at question answering and retrieval-augmented workflows (Liu et al., 2024). These models are adapted for dialogue-oriented reasoning and factuality under supervision and preference data. The 8B and 70B sizes provide options trading latency for accuracy. Refer to Table 10 for sizes and costs.

---

[3]https://www.maa.org/math-competitions

**Gemma-2 (9B)**. Gemma-2 is Google's second-generation open family that introduces architectural refinements for practical-size models while advancing reasoning and multilingual performance (Team et al., 2024b). The 9B variant is a commonly adopted middle ground between capability and deployability. It serves as a base for domain-tuned and coding-specialized derivatives. Costs are summarized in Table 10.

**Mistral-Nemo (12B)**. Mistral-Nemo is a collaboratively developed open-weight model emphasizing efficient inference and high-quality instruction following (Mistral AI, 2024). With 12B parameters, it targets general-purpose chat, reasoning, and code assistance while remaining deployment-friendly. It is often used on NVIDIA accelerators and associated toolchains. See Table 10 for pricing.

**LLaMA-3.3 Nemotron Super (49B)**. Nemotron Super (49B) represents an instruction-tuned assistant model associated with NVIDIA's Nemotron lineup and preference-optimization tooling (Wang et al., 2024b). It emphasizes helpfulness, safety, and strong reasoning via high-quality preference data. Positioned between lightweight and frontier models, it seeks strong accuracy with manageable cost. Pricing appears in Table 10.

**LLaMA-3.1 Nemotron (51B)**. The 51B Nemotron variant builds on the LLaMA-3.1 family with large-scale instruction tuning and preference modeling for chat and tool-use scenarios (Wang et al., 2024b). It aims to combine robust knowledge with alignment for reliable multi-turn QA. This size targets improved quality over small/medium models while controlling inference cost. See Table 10.

**Mixtral (8×7B)**. Mixtral 8×7B is a sparse Mixture-of-Experts (MoE) model where a small subset of experts is activated per token, delivering strong performance at efficient compute (Jiang et al., 2024). It inherits the Mistral architecture and uses routing to select experts dynamically, improving scaling characteristics. Widely adopted instruct variants make it a strong all-around choice. Costs are listed in Table 10.

**Mixtral (8×22B)**. Mixtral 8×22B scales the MoE design to larger experts for higher accuracy while retaining the sparse-activation efficiency benefits (Jiang et al., 2024). It is frequently used for multilingual, reasoning, and coding workloads with long inputs. Instruct-tuned releases are popular for production chat systems. Pricing is shown in Table 10.

# E  EXPERIMENTS ON NEW AGENTIC ROLES

Table 11: **Comparison on the setting of new agentic roles**. This table compares `GraphPlanner` under three settings involving two newly added agentic roles (Thinker and Verifier). *New-role-train* evaluates whether `GraphPlanner` can learn to use the new roles through RL training. *New-role-zero-shot* tests the zero-shot generalization ability of `GraphPlanner` to new roles without any role-specific training. *New-role-few-shot* examines few-shot generalization by providing limited historical interactions involving the new roles during testing.

| Setting | Math | Code | CS | WK | Popular |
|---|---|---|---|---|---|
| GraphPlanner | 67.0% | 76.0% | 78.0% | 38.0% | 52.0% |
| New-role-zero-shot | 68.5% | 77.0% | 78.3% | 38.5% | 52.2% |
| New-role-few-shot | 69.6% | 77.8% | 78.8% | 39.0% | 52.4% |
| New-role-train | **70.5%** | **78.5%** | **79.0%** | **39.5%** | **52.5%** |

To explore whether `GraphPlanner` can adapt and generalize to new agentic roles, we introduce two additional agent roles that are widely used—Thinker (Wei et al., 2022; Wang et al., 2022; Chen et al., 2023b) and Verifier (Lightman et al., 2023; Zhang et al., 2024b; Setlur et al., 2024)—on top of the original three. The Thinker agent processes input queries through systematic reasoning to produce detailed draft analyses, with its prompt template and role description provided in Tables 22 and 27 of the Appendix, respectively. Similarly, the Verifier agent evaluates the accuracy and quality of the generated content before the final output, and its prompt template and role description are also detailed in Tables 23 and 28 of the Appendix. We further design three ablation variants to compare with `GraphPlanner` using only three agentic roles:

- **New-role-train**: This setting is designed to evaluate whether `GraphPlanner` can learn to use new agentic roles through RL training. Specifically, this setting extends the original

`GraphPlanner` by adding two additional agentic roles, the *Thinker* and *Verifier*, and trains and tests the model on the tasks and LLMs used in the Phase-2 setting.

- **New-role-zero-shot**: This setup aims to examine the zero-shot generalization ability of `GraphPlanner` to new roles. Therefore, the training stage is identical to the current version of `GraphPlanner`, whereas during testing the policy is allowed to choose among five agentic roles.
- **New-role-few-shot**: This setting also evaluates the few-shot generalization ability of `GraphPlanner` to new roles. In this case, the training stage remains the same as in the original `GraphPlanner`, but during testing we augment $\mathcal{G}_{history}$ with 50 historical interactions generated by randomly selecting 1% of the training queries and pairing them with all five agentic roles. During testing, the policy can again choose among all five agentic roles.

We report our results in Table 11. Comparing the New-role-train setting with the original `GraphPlanner` shows that incorporating additional agentic roles leads to consistent performance improvements, indicating that `GraphPlanner` is highly adaptable to different agent-role configurations. Furthermore, the results from the New-role-zero-shot and New-role-few-shot settings reveal that `GraphPlanner` generalizes effectively to previously unseen agentic roles: even without role-specific training, it can quickly adapt to new agentic setups and achieve higher task performance. Overall, these findings demonstrate that `GraphPlanner` not only learns to leverage new roles when trained on them, but also possesses strong zero-shot and few-shot generalization capabilities across diverse agentic-role environments.

## F  ADDITIONAL ABLATIONS ON OTHER GRAPH ENCODERS

Table 12: **Ablation on different graph encoders within the `GraphPlanner` architecture across five scenarios.** We replace our proposed encoder `GARNet` with two commonly used alternatives: (1) a GAT-based encoder and (2) a GraphTransformer-based encoder, as detailed in the footnotes below. This comparison evaluates whether the performance gains of `GraphPlanner` stem from the design of our lightweight graph encoder.

| Setting | Math | Code | CS | WK | Popular |
|---|---|---|---|---|---|
| GAT | 0.643 | 0.739 | 0.756 | 0.358 | 0.493 |
| GraphTransformer | 0.647 | 0.743 | 0.759 | 0.353 | 0.491 |
| GraphPlanner | **0.670** | **0.760** | **0.780** | **0.380** | **0.520** |

In this section, we perform ablations on different graph encoders to verify the effectiveness of `GARNet`. We design two ablation variants to compare with `GraphPlanner`:

- **GAT**: Replaces `GARNet` with a graph attention network[4].
- **GraphTransformer**: Replaces `GARNet` with a GraphTransformer[5].

We report our results in Table 12. Across all five scenarios, `GraphPlanner` achieves the strongest overall performance compared with both GAT and GraphTransformer. Relative to GAT, it yields consistent gains ranging from **2.8% to 6.1%** across domains, with the largest improvements observed in the WK and Popular scenarios. Compared with the heavier GraphTransformer, `GraphPlanner` still provides **2.3%–7.6%** relative improvements while maintaining substantially lower architectural and computational overhead. These results confirm that our tailored graph encoder `GARNet` not only delivers the best accuracy but also remains a lightweight and efficient alternative to transformer-based graph encoders.

## G  ADDITIONAL ABLATIONS ON HISTORICAL INFORMATION PROCESSING

In this section, we compare `GraphPlanner` with other methods that use LLM to process historical interactions and make routing decisions. To be specific, we design two baselines based on Router-R1:

---

[4]https://pytorch-geometric.readthedocs.io/en/2.4.0/generated/torch_geometric.nn.models.GAT.html

[5]https://pytorch-geometric.readthedocs.io/en/latest/tutorial/graph_transformer.html

Table 13: **Ablations on historical information processing.** We compare different strategies for incorporating past interaction histories into the routing process. The *History-summary* setting stores all prior interactions and injects a summary of the histories most relevant to the current query, constrained by a 32,768-token context limit. The *History-retrieval* setting augments this approach by retrieving the top $K = 5$ most similar interaction histories and inserting the retrieved contexts directly into the routing prompt. This ablation evaluates how summary-based versus retrieval-based historical information influences routing performance.

| Setting | Math | Code | CS | WK | Popular |
|---|---|---|---|---|---|
| Router-R1 | 0.45 | 0.52 | 0.81 | 0.29 | 0.37 |
| History-retrieval | 0.46 | 0.62 | 0.73 | 0.12 | 0.39 |
| History-summary | 0.51 | 0.62 | 0.75 | 0.14 | 0.36 |
| GraphPlanner | **0.67** | **0.76** | **0.78** | **0.38** | **0.52** |

- **History-summary**: This setting extends Router-R1 by explicitly storing all past interaction histories. During both training and testing, the Router-R1 base model summarizes the interaction histories most relevant to the current query, constrained by its maximum context length (32,768 tokens). The resulting summary is then injected into the routing prompt as additional contextual guidance for selecting the appropriate route.
- **History-retrieval**: Compared with the *History-summary* setting, which only incorporates summaries of neighboring interactions, this variant additionally performs retrieval: it identifies the top K=5 most similar interaction histories to the current query and incorporates these retrieved contexts directly into the routing prompt. This design allows the router to leverage both global historical summaries and fine-grained retrieved evidence when making routing decisions.

We report our results in Table 13. We can observe that LLM-based historical information processing provides only limited improvements over Router-R1, as both summarization and retrieval must operate over highly heterogeneous and unstructured interaction histories. Such histories contain mixed-quality reasoning traces and contextually entangled signals that LLMs struggle to exploit when injected directly as summaries or retrieved text. In contrast, GraphPlanner achieves substantial and consistent gains across all scenarios, outperforming the best history-based method by **31.4%** on Math, **22.6%** on Code, **4.0%** on CS, **171.4%** on WK, and **33.3%** on Popular. These results demonstrate that our graph-based encoder GARNet provides a significantly more principled and robust mechanism for modeling complex historical interactions, enabling GraphPlanner to capture cross-interaction structure, suppress noise, and effectively propagate useful relational information for routing.

# H  ADDITIONAL EXPERIMENTS ON NEW DATASET

Table 14: **Performance on unseen dataset AIME in Phase-2. Bold** and underline indicate the best and second-best results.

| Metric | Single-round Router | | | | | Multi-round Router | | | | GraphPlanner |
|---|---|---|---|---|---|---|---|---|---|---|
| | Router-KNN | Router-MLP | Router-SVM | RouterDC | GraphRouter | Prompt LLM | Router-KNN-MR | R2-Reasoner | Router-R1 | |
| Acc (%) | 3.95 | 7.14 | 4.43 | 2.90 | 7.56 | 3.40 | 3.71 | 7.30 | 5.21 | **14.7** |

We compare the zero-shot generalization ability between GraphPlanner and baselines on AIME[6] under the setting of Phase-2. Specifically, under the Phase-2 setting, we train all methods and conduct zero-shot evaluation on the AIME datasets from 2016 to 2025, whose detailed descriptions are provided in Section D.1. We report our results in Table 14. GraphPlanner achieves the strongest zero-shot performance on the unseen AIME dataset with an accuracy of 14.7%, which is almost twice as high as the best baseline accuracy of 7.56%. Most single-round and multi-round routers remain below 8%, indicating limited transfer ability on competition-level math problems. In contrast, GraphPlanner benefits from its graph-structured workflow planning, which provides substantially better generalization to complex reasoning tasks.

---

[6] https://www.maa.org/math-competitions

## I COMPARISON ON TIME COST

Table 15: **Time cost comparison across all methods in Phase-2.** We report four metrics: *Data Collecting Time*, the cost of generating interaction data; *NN Training Time*, the time required to optimize router parameters; *Total Time for Training*, the sum of both for fair comparison across supervised and RL-based methods; and *Inference Time*, the average latency per query during deployment.

| Metric | Single-round Router | | | | | Multi-round Router | | | | GraphPlanner |
|---|---|---|---|---|---|---|---|---|---|---|
| | Router-KNN | Router-MLP | Router-SVM | RouterDC | GraphRouter | Prompt LLM | Router-KNN-MR | R2-Reasoner | Router-R1 | |
| Data Collecting Time (min) | 395 | 395 | 395 | 395 | 395 | / | / | 0 | 0 | 0 |
| NN Training Time (min) | 5.4 | 8.2 | 7.7 | 11 | 6.2 | / | / | 360 | 300 | 120 |
| Total Time for Training (min) | 400.4 | 403.2 | 402.7 | 406 | 401.2 | / | / | 360 | 300 | **120** |
| Inference Time (s/query) | 2.2 | 2.4 | 2.2 | 2.3 | 2.1 | 10.5 | 9.3 | 8.3 | 3.6 | **1.2** |

We compare the training and inference time costs of all methods under the Phase-2 setting. In the training stage, supervised approaches such as GraphRouter require collecting interaction data between each training query and all LLMs before training begins. In contrast, RL-based methods such as GraphPlanner and Router-R1 collect data dynamically during training. To ensure a fair comparison, we report a unified metric, *Total Time for Training*, which is computed as the sum of *Data Collecting Time* and *NN Training Time*. We report our results in Table 15. Table 15 shows that GraphPlanner achieves the lowest overall time cost among all methods in both training and inference. This efficiency primarily comes from our multi-threaded rollout design, which processes multiple queries in parallel and generates routing interactions simultaneously as illustrated in Section C. As a result, GraphPlanner requires only 120 minutes of total training time, which is substantially lower than RL-based baselines such as R2-Reasoner (360 minutes) and Router-R1 (300 minutes), as well as supervised routers that incur a large up-front data collection cost of 395 minutes. During inference, GraphPlanner also achieves the best latency at 1.2 seconds per query, outperforming both single-round and multi-round routers. Finally, because GraphPlanner collects and uses interaction data on-the-fly in an RL fashion, it avoids the expensive full-sweep data generation required by supervised methods, leading to much higher data efficiency as introduced in Section 4.1. Overall, the results demonstrate that GraphPlanner is both computationally efficient and data-efficient across the entire training and inference pipeline.

## J ILLUSTRATIVE EXAMPLES OF GRAPHPLANNER

This section presents illustrative examples of how GraphPlanner generates adaptive workflows in Phase-2 across different task types. As shown in Figure 6, GraphPlanner constructs distinct workflow structures depending on the complexity of the input query. Table 16 details a multi-stage math reasoning example in which the planner decomposes the query into several sub-queries, assigns them to appropriate executors, and then uses a summarizer to integrate intermediate results. Table 17 demonstrates a more complex code-generation case involving nested planning and hierarchical decomposition. In contrast, Table 18 illustrates a direct executor-only path for simple natural-language questions. Together, these examples highlight GraphPlanner's ability to flexibly choose between single-step execution, multi-step reasoning, and hierarchical planning based on task difficulty and structure.

## K PROMPT USAGE

## L THE USE OF LARGE LANGUAGE MODELS (LLMS)

During the preparation of this manuscript, we used an LLM to assist with improving the readability of the text. The tool was employed exclusively for grammar correction, sentence restructuring, and minor stylistic refinements. All substantive intellectual contributions, including research design, analysis, and conclusions, were produced independently by the authors.

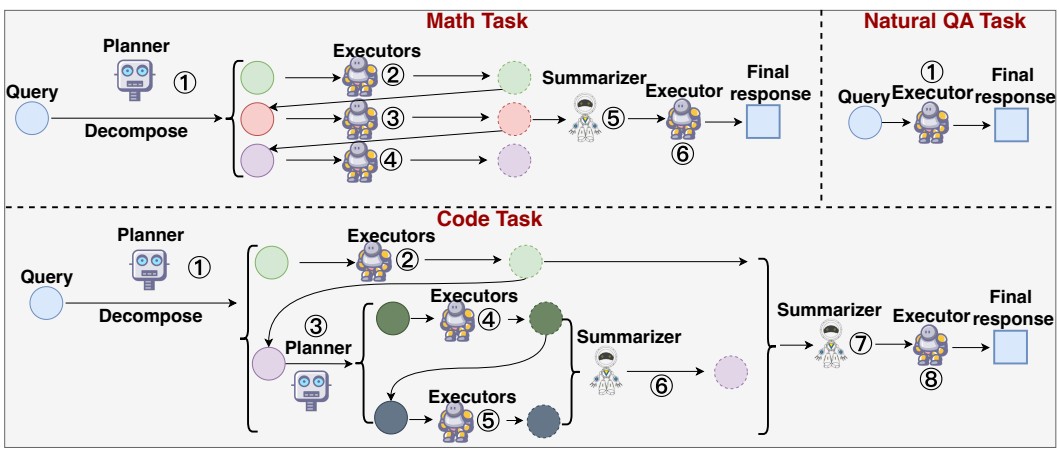

Figure 6: **Illustrative examples of `GraphPlanner`'s workflow generations in phase-2.** The figure illustrates how GraphPlanner adaptively constructs different workflow paths for different task types. For math tasks, the planner first decomposes the query into several sub-queries. Each sub-query is processed by an executor, and the intermediate results are then merged by a summarizer before a final executor produces the answer. For code tasks, GraphPlanner performs nested planning: the initial decomposition triggers a second planning step for one of the sub-queries, producing deeper sub-components that are executed and then summarized into multiple layers of reasoning before producing the final output. For simple natural-language questions, GraphPlanner identifies that no decomposition is needed and directly selects an executor-only path to produce the final response in a single step. Overall, the diagram shows that GraphPlanner flexibly adapts workflow depth and structure based on task complexity, ranging from single-step execution to multi-stage and hierarchical planning.

Table 16: **Phase-2 `GraphPlanner` workflow example for a math QA task.** This example illustrates the full Phase-2 `GraphPlanner` workflow, where the planner decomposes the original question into multiple sub-queries and assigns them to different LLMs. Executors solve each sub-query individually, and intermediate results are summarized by a dedicated summarizer to form a coherent reasoning context. Finally, another executor uses this structured context to compute the final answer. The workflow demonstrates the multi-stage architecture of `GraphPlanner`, including sub-query decomposition, parallel role assignment, structured summarization, and cross-model collaboration.

| Step | Query / Sub-query | LLM | Role | Response (Summary) | Acc | Cost |
|------|-------------------|-----|------|--------------------|-----|------|
| 1 | Original question: "A notebook costs \$3 and a pen costs \$2. What is the total cost of 4 notebooks and 3 pens?" | LLaMA-3.1-8B | Planner | Produces three sub-queries: (Q1) notebook subtotal; (Q2) pen subtotal; (Q3) prepare combined calculation. | 0 | 110 |
| 2 | Q1: "Compute the subtotal for 4 notebooks at \$3 each." | Qwen2.5-7B | Executor | Computes $4 \times 3 = 12$. | 0 | 95 |
| 3 | Q2: "Compute the subtotal for 3 pens at \$2 each." | Gemma-2-9B | Executor | Computes $3 \times 2 = 6$. | 0 | 125 |
| 4 | Q3: "Using the subtotals (\$12 and \$6), prepare the combined information for final reasoning." | LLaMA-3.1-8B | Executor | Returns structured summary: "Notebook subtotal = 12; Pen subtotal = 6." | 0 | 140 |
| 5 | "Summarize all intermediate results into a concise final reasoning context." | LLaMA-3-70B-ChatQA | Summarizer | Generates final reasoning context: "Total = 12 + 6." | 0 | 480 |
| 6 | Final answer step: "Given the original question and the summarized reasoning (12 + 6), compute the final result." | Qwen2.5-14B | Executor | Outputs: "The total cost is \$18." | 1 | 260 |

Table 17: **Phase-2 `GraphPlanner` Workflow Example for a Code Task.** This example demonstrates the nested planning structure of the Phase-2 `GraphPlanner` workflow for a code QA task. The planner first decomposes the original programming problem into two high-level sub-queries. After the executors answer these, `GraphPlanner` triggers a second round of planning, where the planner further breaks down one sub-query into two finer-grained sub-questions about filtering logic and output-string construction. Executors then handle each low-level sub-query independently. The summarizer consolidates both layers of intermediate reasoning into a unified implementation plan, which is combined again with earlier results to form the final structured context. Finally, an executor produces the complete Python function. This workflow illustrates how `GraphPlanner` supports hierarchical decomposition, multi-stage reasoning, and coordinated collaboration across multiple LLMs.

| Step | Query / Sub-query | LLM | Role | Response (Summary) | Acc | Cost |
|---|---|---|---|---|---|---|
| 1 | Original task: "Implement `remove_digits(s: str) -> str`." | LLaMA-3.1-8B | Planner | Produces two sub-queries: (Q1) describe digit-removal rule; (Q2) outline implementation steps. | 0 | 120 |
| 2 | Q1: "Describe the rule for removing digits from a string." | Qwen2.5-7B | Executor | Returns rule: iterate over characters and keep only non-digit characters. | 0 | 95 |
| 3 | Q2: "Outline the implementation steps for `remove_digits`." | LLaMA-3.1-8B | Planner | Decomposes into: (Q2a) describe filtering logic; (Q2b) describe final string construction. | 0 | 115 |
| 4 | Q2a: "Describe filtering logic for keeping non-digit characters." | CodeGemma-7B | Executor | Explains: check each character with `not ch.isdigit()`. | 0 | 200 |
| 5 | Q2b: "Describe how to construct the final output string." | Qwen2.5-7B | Executor | Describes: collect filtered characters and join them into a new string. | 0 | 95 |
| 6 | "Summarize Q2a and Q2b into a unified implementation plan." | LLaMA-3-70B-ChatQA | Summarizer | Produces concise plan: filter non-digit characters → join into result. | 0 | 480 |
| 7 | "Combine Q1's rule with the summarized plan from Step 6." | LLaMA-3-70B-ChatQA | Summarizer | Merged reasoning: removal rule + full implementation steps. | 0 | 480 |
| 8 | Final answer: "Using the initial query and merged summary, produce the final Python function." | Qwen2.5-14B | Executor | Returns final function: "remove_digits". | 1 | 260 |

Table 18: **Phase-2 `GraphPlanner` Workflow Example for a Natural QA Task.** This example illustrates the simplest execution path within the Phase-2 `GraphPlanner` workflow. For straightforward natural-language questions that require no decomposition or multi-stage reasoning, the planner selects a direct executor-only route. The chosen LLM receives the original query and immediately produces the final answer without invoking additional planners or summarizers. This demonstrates `GraphPlanner`'s ability to adaptively determine when complex workflow construction is unnecessary and efficiently route easy queries through a single-step executor path.

| Step | Query | LLM | Role | Response (Summary) | Acc | Cost |
|---|---|---|---|---|---|---|
| 1 | "Who painted the Mona Lisa?" | Qwen2.5-14B | Executor | "The Mona Lisa was painted by Leonardo da Vinci." | 1 | 85 |

Table 19: **Planner prompt template for sub-query decomposition**.

You are a query decomposition assistant. Your task is to decompose the user's query into atomic and independent sub-queries.
**Inputs:** - Original query: {QUERY} - Parent queries: {PARENT_QUERIES} - Previous sibling responses: {SIBLING_RESPONSES}
**Instructions:** - Determine the optimal number of sub-queries (1–3). - Ensure each sub-query is self-contained and non-overlapping. - Avoid redundancy by considering {SIBLING_RESPONSES}. - Adjust the number of sub-queries depending on complexity.
**Output format:** - List 1–3 sub-queries. - One sub-query per line. - No numbering or extra commentary.

Table 20: **Executor prompt template for query answering**.

You are a helpful assistant. Answer the given (sub-)query with support from full context.
**Inputs:** - Current sub-query: {QUERY} - Original query: {ROOT_QUERY} - Parent queries: {PARENT_QUERIES} - Previous sibling responses: {SIBLING_RESPONSES} - If final execution: summary of sub-query responses {SUMMARY}
**Instructions:** - Interpret the sub-query with reference to full context. - Align the answer with prior responses to ensure consistency. - If this is the final step, synthesize everything into a complete final answer.
**Output format:** - Direct, complete answer in the format required by the task. - No extra commentary.

Table 21: **Summarizer prompt template for parent query synthesis**.

You are a professional summarizer. Your task is to synthesize multiple child answers into a coherent response to the parent query.
**Inputs:** - Parent query: {PARENT_QUERY} - Child answers: {CHILD_ANSWERS}
**Instructions:** - Combine all child answers into a complete, coherent response. - Preserve all important details. - Resolve overlap or conflicts among child answers. - Ensure the response directly addresses {PARENT_QUERY}.
**Output format:** - A single, well-structured paragraph answering the parent query.

Table 22: **Thinker prompt template for sub-query reasoning**.

You are a Thinker Agent in a multi-agent workflow system. Your task is to generate detailed reasoning responses for sub-queries. **Inputs:** - Sub-query: {SUB_QUERY} - Original query: {ROOT_QUERY} - Parent queries: {PARENT_QUERIES} - Previous sibling responses: {SIBLING_RESPONSES} **Instructions:** - Understand the sub-query in context of the original query. - Think step-by-step through the problem with detailed reasoning. - Consider information from previous sibling responses to maintain consistency. - Show your reasoning process clearly before reaching conclusion. **Output format:** - **Reasoning Steps:** List numbered reasoning steps. - **Draft Answer:** Your reasoned response to the sub-query. - Be thorough as your response will be verified by a Verifier Agent.

Table 23: **Verifier prompt template for response verification**.

You are a Verifier Agent in a multi-agent workflow system. Your task is to verify draft responses and produce refined, verified outputs. **Inputs:** - Sub-query: {SUB_QUERY} - Original query: {ROOT_QUERY} - Draft response from Thinker: {DRAFT_RESPONSE} - Previous sibling verified responses: {VERIFIED_SIBLING_RESPONSES} **Instructions:** - Verify accuracy, completeness, consistency, and logical soundness. - If draft response is correct: approve and format cleanly. - If issues found: correct errors and improve the response. - Ensure consistency with other verified sibling responses. **Output format:** - **Verification Result:** [APPROVED/REVISED] - **Issues Found:** List specific problems identified (if any). - **Verified Response:** Final verified answer to the sub-query.

Table 24: **Description of Planner agent**.

The Planner acts as a decomposition agent. Its primary role is to analyze a complex user query and break it down into a set of clear, atomic sub-questions that can be addressed independently. This ensures that each sub-query targets a specific aspect of the original request, reducing ambiguity and overlap. The Planner helps streamline multi-step reasoning or multi-part queries by structuring them into manageable components for downstream processing.

Table 25: **Description of Executor agent**.

The Executor serves as the answering agent. It is responsible for generating responses to the user's queries, either directly or by incorporating additional background context when necessary. When context is provided, the Executor uses it to produce a more informed and grounded response. It can operate in both raw query execution mode or in a final, context-aware answering mode, depending on the task's stage and goal.

Table 26: **Description of Summarizer agent**.

The Summarizer functions as the condensation agent. Its role is to distill long or complex content into a concise, coherent, and fluent summary. Instead of listing key points, the Summarizer rewrites the original input into a well-structured passage that captures the essential meaning, making the information easier to digest and understand at a glance.

Table 27: **Description of Thinker agent**.

The Thinker acts as a reasoning agent. Its primary role is to process sub-queries through systematic step-by-step analysis, generating detailed reasoning chains that lead to well-founded conclusions. The Thinker excels at breaking down complex problems into logical steps, exploring multiple approaches, and providing comprehensive draft responses with explicit reasoning processes. It serves as the core analytical component that transforms sub-queries into thoroughly reasoned draft answers for subsequent verification.

Table 28: **Description of Verifier agent**.

The Verifier serves as a quality assurance agent. It is responsible for critically evaluating draft responses from Thinker agents, checking for accuracy, completeness, logical consistency, and alignment with the original query context. The Verifier can identify factual errors, reasoning flaws, or gaps in responses, and either approve correct drafts or provide refined corrections. It acts as the final quality gate, ensuring that only verified, high-confidence responses proceed to the synthesis stage.

Table 29: **Description of Qwen2.5 (7b)**.

Qwen2.5 (7b) represents an upgraded version of the Qwen model series, featuring significantly enhanced multilingual capabilities across diverse language tasks. This improved model offers excellent value at $0.20 per million input tokens and $0.20 per million output tokens.

Table 30: **Description of CodeGemma (7b)**.

CodeGemma (7b) is a specialized variant of the Gemma model family that focuses exclusively on code generation and completion tasks. This programming-oriented model provides robust coding assistance capabilities at an affordable rate of $0.20 per million input tokens and $0.20 per million output tokens.

Table 31: **Description of Mistral (7b)**.

Mistral (7b) is a highly efficient open-weight model with 7 billion parameters, optimized for fast inference and strong performance on general text generation tasks. It offers competitive pricing at $0.20 per million input tokens and $0.20 per million output tokens.

Table 32: **Description of LLaMA-3.1 (8b)**.

LLaMA-3.1 (8b) is Meta's 8-billion parameter model from the advanced Llama-3 series, specifically designed for conversational AI and complex reasoning tasks. This versatile model combines strong performance with reasonable costs at $0.20 per million input tokens and $0.20 per million output tokens.

Table 33: **Description of LLaMA-3 ChatQA (8b)**.

LLaMA-3 ChatQA (8b) is an NVIDIA fine-tuned 8-billion parameter model specifically optimized for question-answering and reasoning applications. This specialized model delivers enhanced performance in conversational AI scenarios at $0.20 per million input and output tokens.

Table 34: **Description of Gemma-2 (9b)**.

Gemma-2 (9b) is a 9-billion parameter instruction-tuned model from Google, designed for general text processing and conversational applications. This compact yet capable model offers exceptional value with ultra-low pricing of $0.10 per million input tokens and $0.10 per million output tokens.

Table 35: **Description of Mistral-Nemo (12b)**.

Mistral-Nemo (12b) is a 12-billion parameter model that combines innovative Mistral architecture with NeMo technology for enhanced performance. This hybrid approach delivers superior capabilities across various tasks, priced at $0.30 per million input tokens and $0.30 per million output tokens.

Table 36: **Description of LLaMA-3.3 Nemotron Super (49b)**.

LLaMA-3.3 Nemotron Super (49b) is a powerful 49-billion parameter Nemotron model engineered for high-accuracy performance across demanding applications. This advanced model delivers exceptional results for complex tasks, available at $0.90 per million input and output tokens.

Table 37: **Description of LLaMA-3.1 Nemotron (51b)**.

LLaMA-3.1 Nemotron (51b) is NVIDIA's 51-billion parameter alignment model that focuses on producing safe, helpful, and accurate responses. This enterprise-grade model emphasizes responsible AI deployment and is priced at $0.90 per million input and output tokens.

Table 38: **Description of Mixtral (8x7b)**.

Mixtral (8x7b) is a 56-billion parameter Mixture of Experts (MoE) model composed of eight 7-billion parameter expert models, specifically optimized for creative text generation. This innovative architecture provides high-quality outputs while maintaining efficiency, available at $0.60 per million input and output tokens.

Table 39: **Description of LLaMA-3 ChatQA (70b)**.

LLaMA-3 ChatQA (70b) is a 70-billion parameter model specifically optimized for conversational AI and chat applications. This large-scale model provides sophisticated dialogue capabilities and nuanced understanding, available at $0.90 per million input and output tokens.

Table 40: **Description of Mixtral (8x22b)**.

Mixtral (8x22b) is an advanced 176-billion parameter Mixture of Experts model comprising eight 22-billion parameter expert components. This large-scale MoE architecture delivers exceptional performance across diverse tasks while maintaining computational efficiency, priced at $1.20 per million input and output tokens.

| LLM Role | Function |
| --- | --- |
| Planner | Decomposes a complex query into atomic sub-queries and organizes the workflow. |
| Executor | Generates answers for sub-queries with or without contextual grounding. |
| Summarizer | Aggregates multiple intermediate outputs into a coherent final response. |
| Thinker | Performs systematic reasoning to produce detailed draft analyses before execution. |
| Verifier | Evaluates the correctness and quality of generated content before finalization. |

Table 41: LLM roles used in GraphPlanner, including additional agentic roles introduced in the appendix.

