# OpenReview forum: "GraphPlanner: Graph Memory-Augmented Agentic Routing for Multi-Agent LLMs"
_ICLR.cc/2026/Conference — ICLR 2026 Poster_

### Official Review · Reviewer_SiqR · 2025-10-17

**Soundness:** 3
**Presentation:** 3
**Contribution:** 3
**Rating:** 8
**Confidence:** 3

**Summary:**

This paper introduces a novel LLM routing paradigm, an agentic router setting. Unlike traditional routers, the proposed GraphPlanner not only selects the backbone LLM but also assigns specific agentic roles, e.g., planner, executor, or summarizer, to solve the initial query. The router is parameterized by GARNet, a graph-based model that captures the relationships between queries, agentic roles, and responses. GARNet is optimized using PPO with a joint loss that balances task-specific performance and computational costs, enabling adaptive router learning. Extensive experiments across 14 diverse tasks, spanning in-domain and out-of-domain settings, as well as inductive and transductive scenarios, demonstrate GraphPlanner's SOTA performance, exceptional balance of cost and performance, and strong generalization capabilities.

**Strengths:**

1. The proposed agentic router setting is well-motivated and clearly articulated. While traditional single-round routers are limited to solving isolated queries and multi-round routers can only model sequential workflows, GraphPlanner addresses more complex graph-structured workflows that are under-explored in prior work.

2. GraphPlanner is designed with sound principles, featuring a lightweight implementation and effective contextual history preservation. This ensures that the agentic router is simple to implement and extend. Additionally, the joint optimization of the router, combining task-specific loss and cost constraints, allows the model to learn policies tailored to solved task while maintaining efficiency.

3. The experimental results are extensive, covering a wide range of downstream tasks in both in-domain and out-of-domain scenarios. The authors also include detailed cost-performance trade-offs and further analysis in inductive and transductive settings. These results highlight GraphPlanner's SOTA performance compared to single-round and multi-round routers, its robustness in inductive and out-of-domain tasks, and its favorable balance between performance and computational costs.

4. The paper is well-written, with vivid figures and tables that significantly enhance comprehension.

**Weaknesses:**

1. The description of Phase 1 optimization is somewhat unclear. For instance, in the case of Depth = 1 and Width = 3, does this imply a fixed agentic workflow where the first step always involves an initial planner role, followed by two roles (e.g., summarizer or executor)? Further clarification of this process would be helpful.

2. The paper lacks illustrative examples showing how the router assigns roles to specific models to successfully solve queries. Including such examples would better demonstrate how GraphPlanner operates in practice.

3. The experimented datasets appear relatively simple, as single-round routers already achieve good performance in some scenarios. It would be valuable to test GraphPlanner on more challenging benchmarks, such as agent-related benchmarks, which involve complex workflows and are better suited for role decomposition.

**Questions:**

Please refer to Weaknesses part

---

> ### Author Response · Authors · 2025-11-21
> **Response to Reviewer SiqR**
>
> **Q1. The description of Phase 1 optimization is somewhat unclear. For instance, in the case of Depth = 1 and Width = 3, does this imply a fixed agentic workflow where the first step always involves an initial planner role, followed by two roles (e.g., summarizer or executor)? Further clarification of this process would be helpful.**
>
> **Response:** Thanks for your thoughtful feedback. Your understanding of Phase 1 is correct. Following your suggestion, we have added **Figure 3** and updated the description in **lines 350–353** of the revised PDF to make the Phase-1 setting clearer.
>
> ---
>
> **Q2. The paper lacks illustrative examples showing how the router assigns roles to specific models to successfully solve queries. Including such examples would better demonstrate how GraphPlanner operates in practice.**
>
> **Response:** Thanks for your insightful feedback. Following your suggestion, we have added **Figure 6** and **Tables 16–18** to **Section J of the Appendix** in the revised paper. These additions provide concrete, step-by-step illustrations of how GraphPlanner constructs adaptive multi-agent workflows in Phase 2, along with detailed examples showing how the router assigns appropriate roles and model backbones for representative task types. We believe these visualizations make the end-to-end routing and workflow generation process much clearer.
>
> ---
>
> **Q3. The experimented datasets appear relatively simple, as single-round routers already achieve good performance in some scenarios. It would be valuable to test GraphPlanner on more challenging benchmarks, such as agent-related benchmarks, which involve complex workflows and are better suited for role decomposition.**
>
> **Response:** Thanks for your constructive feedback. In fact, our paper already evaluates LLM routing on several challenging and widely used agent benchmarks, such as LogicGrid  and CommonGen from AgentVerse (Chen et al., 2023a), as shown in Table 5. Following the reviewer’s suggestion, we further include experiments on a highly challenging benchmark, AIME [1].
>
> Specifically, we compare the zero-shot generalization ability of GraphPlanner and the baselines on AIME under the Phase-2 setting. Under this setup, all methods are trained in Phase-2 and then evaluated in a zero-shot manner on the AIME datasets from 2016 to 2025, whose detailed descriptions are provided in Section D.1. The results are reported in the following table.
>
> GraphPlanner achieves the strongest zero-shot performance on the unseen AIME dataset with an accuracy of **14.7%**, which is almost twice as high as the best baseline at **7.56%**. Most single-round and multi-round routers remain below **8%**, indicating limited transferability to competition-level mathematical reasoning problems. In contrast, GraphPlanner benefits from its **graph-structured workflow planning**, which enables substantially better generalization to such complex reasoning tasks. All the above experiments and discussions have been incorporated into **Section H of the Appendix** in the revised PDF.
>
> **Table: Performance on unseen dataset AIME in Phase-2**
>
> | Metric | Router-KNN | Router-MLP | Router-SVM | RouterDC | GraphRouter | Prompt LLM | Router-KNN-MR | R2-Reasoner | Router-R1 | GraphPlanner |
> |--------|-----------|-----------|-----------|---------|------------|-----------|--------------|-----------|---------|--------|
> | **Acc (%)** | 3.95 | 7.14 | 4.43 | 2.90 | 7.56| 3.40 | 3.71 | 7.30 | 5.21 | **14.7** |
>
>  **[1]** https://www.maa.org/math-competitions

---

### Official Review · Reviewer_3pJs · 2025-10-28

**Soundness:** 3
**Presentation:** 2
**Contribution:** 2
**Rating:** 4
**Confidence:** 3

**Summary:**

In this paper, the authors proposed the GraphPlanner, which is a graph-based LLM router for agentic LLM flow prediction. Given a constructed workflow and history graph, the GraphPlanner predicts the best agent role and the LLM for the next step. by training the GNN model with PPO, it can generalize to unseen tasks and models.

**Strengths:**

- The proposed methods show significant improvement over baseline methods under different scenarios.
- The authors provide a comprehensive ablation study to prove the effectiveness of the proposed method.

**Weaknesses:**

- The writing can be improved. In particular, I am pretty confused about the graph construction part.  Why are all nodes connected to the role hub node? Is there any particular consideration behind this design? For multi-round routing, will there be multiple role hub nodes? How are different rounds connected in the graph? Will the role hub node encode the role information of both the history graph and the workflow graph? I believe a more detailed illustration or figure is needed to allow the reader to better understand it.
- For the agent role, the Graphplanner defined three different roles.  I am wondering what the rationale behind it is, and I am curious about whether the proposed methods can generalize to unseen roles after training.
- What's the time cost for GraphPlanner compared to other baselines under different settings? What is the training time for GraphPlanner and other baselines?
- The major contribution and the source of performance improvement come from the history graph. I am wondering, is it necessary to construct historical information into a graph? What if I simply describe all historical information to a (small )LLM and use the same training pipeline to optimize it?

**Questions:**

See above?

---

> ### Author Response · Authors · 2025-11-21
> **Response to Reviewer 3pJs (1/3)**
>
> **Q1. The writing can be improved. In particular, I am pretty confused about the graph construction part. Why are all nodes connected to the role hub node? Is there any particular consideration behind this design? For multi-round routing, will there be multiple role hub nodes? How are different rounds connected in the graph? Will the role hub node encode the role information of both the history graph and the workflow graph? I believe a more detailed illustration or figure is needed to allow the reader to better understand it.**
>
> **Response:** Thanks for your insightful questions. The key insight behind GARNet’s graph construction is that the **semantic embeddings of LLM roles are crucial for routing decisions**, yet they are difficult to obtain or model directly. However, these role semantics are implicitly reflected in the accumulated **interaction history**. GARNet therefore models the interaction histories as a **historical graph** to derive LLM-role semantic embeddings, while also using these embeddings to guide GraphPlanner’s routing decisions over the workflow graph.
>
> Based on this motivation, as is mentioned in Section 3.2, we designate all the **LLM-role nodes as hub nodes**, rather than defining a single hub node. Because an LLM role’s problem-solving capability is an inherent and internalized property—one that becomes increasingly reflected in the interaction history—we connect **all nodes to their corresponding role hub node**. As the interaction history grows, this design enables GARNet to capture increasingly accurate LLM-role semantic embeddings.
>
> Multi-round routing does not introduce any additional role nodes. All newly generated sub-queries or responses in later rounds are appended to the workflow graph and connected to the **same shared role hub nodes**. This implicitly links different rounds through shared neighbors, rather than explicit temporal edges, allowing GARNet to leverage accumulated knowledge throughout multi-step routing. In this way, each role hub node jointly encodes the role-specific information from both the **history graph** and the **workflow graph**, providing a unified semantic representation for routing.
>
> We thank the reviewer again for the valuable suggestion. To make the graph construction component clearer, we followed the recommendation and added additional explanatory details in **Section 3.2** of our revised PDF. We also updated **Figure 2** to present the GraphPlanner graph construction process in a more transparent and intuitive manner.
>
> ---
>
> **Q2. For the agent role, the Graphplanner defined three different roles. I am wondering what the rationale behind it is, and I am curious about whether the proposed methods can generalize to unseen roles after training.**
>
> **Response:**  Thank you for the thoughtful feedback. We provide clarifications regarding the agent roles and the generalization ability of GraphPlanner.
>
> GraphPlanner is a highly flexible agentic routing framework capable of adapting to a wide range of agentic roles. As stated in **lines 83–86**, we adopt the Planner, Executor, and Summarizer profiles because they capture the core functional patterns that commonly appear in agentic workflows and are widely used in prior studies (Barachini & Stary, 2022; Tran et al., 2025). These three roles form a minimal yet expressive abstraction that allows us to demonstrate, without loss of generality, the general agentic routing capabilities of GraphPlanner.
>
> Following the reviewer’s suggestion, we further evaluate GraphPlanner’s flexibility by introducing two widely used additional agent roles—**Thinker** (Wei et al., 2022; Wang et al., 2022; Chen et al., 2023b) and **Verifier** (Lightman et al., 2023; Zhang et al., 2024b; Setlur et al., 2024)—on top of the original three. The Thinker agent performs systematic reasoning to produce detailed draft analyses, with its prompt template and role description provided in **Tables 22 and 27**. The Verifier agent evaluates the accuracy and quality of generated content before final output, and its prompt template and description appear in **Tables 23 and 28**.
>
> **(continued in the next response box)**

---

> > ### Author Response · Authors · 2025-11-21
> > **Response to Reviewer 3pJs (2/3)**
> >
> > **(This is the continuation of our response to Q2.)**
> >
> > To examine GraphPlanner under the expanded five-role setting, we design three ablation variants comparing the five-role configuration with the original three-role setup:
> >
> > - **New-role-train**: Extends GraphPlanner with Thinker and Verifier and trains the model under the Phase-2 tasks and LLMs.
> > - **New-role-zero-shot**: Keeps the training identical to the original three-role GraphPlanner, while allowing the model to select from all five roles during testing.
> > - **New-role-few-shot**: Uses the original training setup and augments $\mathcal{G}_{history}$ during testing with 50 interactions derived from 1% of training queries paired with all five roles.
> >
> > The results in the table below show that the **New-role-train** variant consistently improves performance, indicating that GraphPlanner can effectively utilize the newly added roles during RL training. The **New-role-zero-shot** and **New-role-few-shot** variants also show clear gains over the original method, demonstrating that GraphPlanner generalizes well to previously unseen roles and adapts to the expanded role set without requiring additional role-specific training. Overall, these results confirm that GraphPlanner is not limited by its initial role definitions and provides strong adaptability, zero-shot generalization, and few-shot generalization across diverse agent-role environments. We summarize these experiments in **Appendix E of our revised PDF**.
> >
> > **Table: Comparison under new agentic role settings**
> >
> > | Setting | Math | Code | CS | WK | Popular |
> > |--------|------|------|-----|-----|---------|
> > | GraphPlanner | 67.0% | 76.0% | 78.0% | 38.0% | 52.0% |
> > | New-role-zero-shot | 68.5% | 77.0% | 78.3% | 38.5% | 52.2% |
> > | New-role-few-shot | 69.6% | 77.8% | 78.8% | 39.0% | 52.4% |
> > | New-role-train | 70.5% | 78.5% | 79.0% | 39.5% | 52.5% |
> >
> > ---
> >
> > **Q3. What's the time cost for GraphPlanner compared to other baselines under different settings? What is the training time for GraphPlanner and other baselines?**
> >
> > **Response:** Thanks for your valuable questions. We compare the training and inference time costs of all methods under the Phase-2 setting. In the training stage, supervised approaches such as GraphRouter must collect interaction data between every training query and all LLMs before training begins, whereas RL-based methods such as GraphPlanner and Router-R1 collect data dynamically during training. To enable a fair comparison, we report a unified metric, **Total Time for Training**, defined as the sum of **Data Collecting Time** and **NN Training Time**.
> >
> > As shown in the following table, GraphPlanner achieves the lowest overall time cost across all methods in both training and inference. This advantage primarily comes from our **multi-threaded rollout design**, which processes multiple queries in parallel and generates routing interactions simultaneously (illustrated in Section C). Consequently, GraphPlanner requires only **120 minutes** of total training time, significantly lower than RL-based baselines such as R2-Reasoner (**360 minutes**) and Router-R1 (**300 minutes**), as well as supervised routers that incur a large up-front data collection cost of **395 minutes**.
> >
> > During inference, GraphPlanner also achieves the best latency at **1.2 seconds per query**, outperforming both single-round and multi-round routers. Moreover, because GraphPlanner collects and utilizes interaction data on-the-fly in an RL fashion, it avoids the expensive full-sweep data generation required by supervised methods, resulting in substantially higher data efficiency (**details in Section 4.1**).
> >
> > Overall, these results demonstrate that GraphPlanner is both **computationally efficient** and **data-efficient** across the entire training and inference pipeline. All the above experiments and discussions have been incorporated into **Section I of the Appendix** in the revised PDF.
> >
> > **Table: Time cost comparison across all methods in phase-2**
> >
> > | Metric | Router-KNN | Router-MLP | Router-SVM | RouterDC | GraphRouter | Prompt LLM | Router-KNN-MR | R2-Reasoner | Router-R1 | GraphPlanner |
> > |--------|-----------|-----------|-----------|---------|------------|-----------|--------------|-----------|---------|--------|
> > | **Data Collecting Time (min)** | 395 | 395 | 395 | 395 | 395 | / | / | 0 | 0 | 0 |
> > | **NN Training Time (min)** | 5.4 | 8.2 | 7.7 | 11 | 6.2 | / | / | 360 | 300 | 120 |
> > | **Total Time for Training (min)** | 400.4 | 403.2 | 402.7 | 406 | 401.2 | / | / | 360 | 300 | **120** |
> > | **Inference Time (s/query)** | 2.2 | 2.4 | 2.2 | 2.3 | 2.1 | 10.5 | 9.3 | 8.3 | 3.6 | **1.2** |

---

> > > ### Author Response · Authors · 2025-11-21
> > > **Response to Reviewer 3pJs (3/3)**
> > >
> > > **Q4. The major contribution and the source of performance improvement come from the history graph. I am wondering, is it necessary to construct historical information into a graph? What if I simply describe all historical information to a (small) LLM and use the same training pipeline to optimize it?**
> > >
> > > **Response:** Thanks for your constructive feedback. We would first like to clarify that the core contribution and performance improvements of our method do not come solely from the history graph. As shown in the ablations in **Section 4.3**, the gains of GraphPlanner arise from the combination of our **RL-based decision-making framework**, the construction of the **workflow graph**, and the **history graph**. All three components are essential to the overall effectiveness of our approach.
> > >
> > > Following the reviewer’s suggestion, we also compare GraphPlanner with methods that rely on LLMs to directly process historical interactions and make routing decisions. Specifically, we design two baselines based on Router-R1, implemented using qwen2.5-3b-instruct:
> > >
> > > - **History-summary**: This setting extends Router-R1 by storing all past interaction histories. During both training and testing, Router-R1 summarizes the interaction histories most relevant to the current query, subject to its maximum context length of 32,768 tokens. The summary is then added to the routing prompt as contextual guidance.
> > >
> > > - **History-retrieval**: Beyond summarization, this setting retrieves the top **K = 5** most similar interaction histories based on the query’s similarity to past queries. The retrieved contexts are directly injected into the routing prompt. This allows the model to use both global summaries and fine-grained retrieved evidence.
> > >
> > > We report the results in the following table. LLM-based historical information processing provides only limited improvements over Router-R1, due to the highly heterogeneous and unstructured nature of interaction histories. These histories include mixed-quality reasoning traces and entangled contextual signals that LLMs struggle to use effectively when injected as raw summaries or retrieval snippets. In contrast, GraphPlanner achieves **substantial and consistent gains** across all scenarios, outperforming the best history-based method by **31.4% on Math**, **22.6% on Code**, **4.0% on CS**, **171.4% on WK**, and **33.3% on Popular**. These results show that our graph-based encoder **GARNet** provides a far more principled and robust mechanism for modeling complex historical interactions. GARNet enables GraphPlanner to capture cross-interaction structure, suppress noise, and propagate useful relational information—leading to significantly stronger routing performance than LLM-based summary or retrieval methods. All the above experiments and discussions have been incorporated into **Section G of the Appendix** in the revised PDF.
> > >
> > > **Table: Ablations on historical information processing**
> > >
> > > | Setting | Math | Code | CS | WK | Popular |
> > > |---------|------|------|-----|-----|---------|
> > > | Router-R1 | 0.45 | 0.52 | 0.81 | 0.29 | 0.37 |
> > > | History-retrieval | 0.46 | 0.62 | 0.73 | 0.12 | 0.39 |
> > > | History-summary | 0.51 | 0.62 | 0.75 | 0.14 | 0.36 |
> > > | **GraphPlanner** | **0.67** | **0.76** | **0.78** | **0.38** | **0.52** |

---

> > > > ### Comment · Reviewer_3pJs · 2025-11-24
> > > >
> > > > I would like to thank the authors for the detailed response. Most of my concerns are solved, and I will raise my score accordingly.

---

> ### Author Response · Authors · 2025-11-25
> **Thank you very much for your constructive feedback—may we kindly ask whether you have any remaining concerns?**
>
> Thank you very much for your encouraging comments. We sincerely appreciate the time and thought you have devoted to reviewing our work. **If there are any aspects you feel could be further refined to strengthen the paper, we would be truly grateful for your guidance, as we are fully committed to improving the work wherever possible.**
>
> **If our revisions satisfactorily address your concerns and meet your expectations, we would be very glad if you might consider reflecting that in your final evaluation—for instance, potentially further raising the score from a 6 to an 8 should you feel the improvements merit it.**
>
> Thank you again for your valuable feedback and kind support.

---

### Official Review · Reviewer_E6qX · 2025-11-01

**Soundness:** 3
**Presentation:** 2
**Contribution:** 2
**Rating:** 4
**Confidence:** 4

**Summary:**

This paper introduces GraphPlanner, a heterogeneous graph-based agentic router that extends LLM routing beyond static or multi-round settings into dynamic multi-agent coordination. The paper formulates workflow generation as a Markov decision process, where at each step both the LLM backbone and the agent role (Planner, Executor, Summarizer) are selected. A novel graph neural network, GARNet, integrates both the current workflow graph and historical interactions, enabling inductive and transductive inference. The proposed method is trained using proximal policy optimization. Experiments across 14 tasks and 6 domains show improved accuracy and reduced GPU cost over prior routers, with strong zero-shot generalization.

**Strengths:**

- **S1.** Novel formulation of LLM routing as a graph-based agentic workflow generation problem using an MDP framework. The reinforcement learning-based optimization adds a dynamic decision-making aspect missing in static routers.

- **S2.** Integration of historical context through GARNet for inductive and transductive inference provides a principled way to leverage past interactions.

- **S3.** Comprehensive evaluation across several tasks and domains. Demonstrated efficiency with substantial reductions in GPU usage and token consumption. Zero-shot generalization to unseen LLMs and tasks shows strong adaptability and robustness.

**Weaknesses:**

- **W1.** The system fixes roles to Planner, Executor, Summarizer, which may constrain scalability to more complex or hierarchical workflows.

- **W2.** The experiments fix depth = 1-2 and width = 2-3, which seems too limited for realistic multi-step tasks.

- **W3.** The paper omits recent agent workflow generation systems (e.g., ADAS, AFlow, AgentSquare), which are directly relevant to the *workflow generation* claim in Table 3.


- **W4.** Insufficient discussion of real-world deployment or integration with tool-based or API-based LLM ecosystems, despite the *agentic* framing.

**Questions:**

- **Q1.** How would GraphPlanner perform when additional agent types (e.g., retriever, verifier) are introduced? Is the policy flexible enough to handle new roles dynamically?

- **Q2.** Why are other graph encoders (e.g., GAT, GraphTransformer) not compared or ablated against GARNet?

- **Q3.** What motivated the use of Longformer embeddings? Was long-context modeling empirically necessary?

- **Q4.** Can the authors clarify what constitutes “existing LLM workflows” in Phase 1 evaluation (Table 1)? Were these synthetic or from real systems?

---

> ### Author Response · Authors · 2025-11-21
> **Response to Reviewer E6qX (1/3)**
>
> **Q1. The system fixes roles to Planner, Executor, Summarizer, which may constrain scalability to more complex or hierarchical workflows. How would GraphPlanner perform when additional agent types (e.g., retriever, verifier) are introduced? Is the policy flexible enough to handle new roles dynamically?**
>
> **Response:**  Thank you for the thoughtful feedback. We would like to clarify that **GraphPlanner is a highly flexible agentic routing framework capable of adapting to a wide range of agentic roles**. As stated in **[lines 83–86]**, we adopt the Planner, Executor, and Summarizer profiles because they capture the core functional patterns that commonly appear in agentic workflows and are widely used in the literature (Barachini & Stary, 2022; Tran et al., 2025). Using these three roles allows us to demonstrate, without loss of generality, the effectiveness of GraphPlanner as an agentic routing framework.
>
> Following the reviewer’s suggestion, we further evaluate GraphPlanner’s flexibility by **introducing two widely used additional agent roles—Thinker (Wei et al., 2022; Wang et al., 2022; Chen et al., 2023b) and Verifier (Lightman et al., 2023; Zhang et al., 2024b; Setlur et al., 2024)—on top of the original three**.  The Thinker agent performs systematic reasoning to generate detailed draft analyses, and its prompt template and role description are provided in Tables 22 and 27.  The Verifier agent evaluates the accuracy and quality of generated content before final output, and its prompt template and description appear in Tables 23 and 28.
>
> To assess how GraphPlanner behaves under this expanded role set, we design three ablation variants comparing the five-role configuration with the original three-role setup:
>
> - **New-role-train**: Extends GraphPlanner with Thinker and Verifier and trains the model under the Phase-2 tasks and LLMs, testing whether GraphPlanner can learn to use new roles through RL training.
> - **New-role-zero-shot**: Keeps training identical to the original three-role GraphPlanner but allows the model to select among all five roles at test time, evaluating zero-shot generalization.
> - **New-role-few-shot**: Retains the original training setup but augments $\mathcal{G}_{history}$ during testing with 50 additional interactions generated by sampling 1% of training queries and pairing them with all five roles, evaluating few-shot generalization.
>
> The results in the following table show that adding new agentic roles in **New-role-train** consistently improves performance, demonstrating that GraphPlanner readily adapts to different role configurations when trained on them.  Moreover, **New-role-zero-shot** and **New-role-few-shot** reveal strong generalization to previously unseen roles: even without role-specific training, GraphPlanner quickly adapts to the expanded role set and improves task performance.  **Overall, these findings confirm that GraphPlanner not only learns to leverage new agentic roles when trained on them, but also exhibits strong zero-shot and few-shot generalization across diverse agent-role environments, reinforcing its flexibility as a general agentic routing framework.** We summarize these experiments and the corresponding content in Appendix E of our revised PDF.
>
> **Table: Comparison on the setting of new agentic roles**
>
> | Setting | Math | Code | CS | WK | Popular |
> |---------|------|------|----|----|---------|
> | GraphPlanner | 67.0% | 76.0% | 78.0% | 38.0% | 52.0% |
> | New-role-zero-shot | 68.5% | 77.0% | 78.3% | 38.5% | 52.2% |
> | New-role-few-shot | 69.6% | 77.8% | 78.8% | 39.0% | 52.4% |
> | New-role-train | **70.5%** | **78.5%** | **79.0%** | **39.5%** | **52.5%** |
>
> ---
>
> **Q2. The experiments fix depth = 1-2 and width = 2-3, which seems too limited for realistic multi-step tasks.**
>
> **Response:**  Thanks for your insightful feedback. As stated in **lines 350–360**, our Phase-1 setting is not restricted to a fixed depth of 1–2 or width of 2–3. These values are chosen only as a representative example to demonstrate that, under a user-defined LLM workflow, GraphPlanner performs more effective routing than alternative routers. Moreover, under the Phase-2 setting, GraphPlanner dynamically generates different agentic workflows for different queries, enabling more efficient LLM routing tailored to the structure of each problem. This adaptive workflow construction allows GraphPlanner to handle realistic multi-step tasks more effectively than fixed-routing approaches.

---

> ### Author Response · Authors · 2025-11-21
> **Response to Reviewer E6qX (2/3)**
>
> **Q3. The paper omits recent agent workflow generation systems (e.g., ADAS, AFlow, AgentSquare), which are directly relevant to the workflow generation claim in Table 3.**
>
> **Response:**  Thanks for your thoughtful question. We would like to clarify that, as illustrated in **lines 35–45 and Section 2**, the core problem addressed in our paper is **LLM routing** in agentic workflows, not the workflow generation task itself. The distinction is fundamental: LLM routing selects different LLM backbones for different queries in order to optimize objectives such as performance or cost, whereas agent workflow generation typically assumes a fixed LLM backbone for all agents and does not involve optimizing the selection of LLM backbones. We introduce agentic roles to extend the applicability of existing LLM routing frameworks to agent workflows. For this reason, both the Introduction and the baseline comparisons focus on representative LLM routing methods. We fully understand and appreciate the reviewer’s suggestion regarding workflow-generation literature. Following this feedback, we have added a discussion in **Additional Related Work (lines 866–887)** of the revised PDF, where we analyze the relationship between existing workflow-generation approaches and our proposed method.
>
> ---
>
> **Q4. Insufficient discussion of real-world deployment or integration with tool-based or API-based LLM ecosystems, despite the agentic framing.**
>
> **Response:**  Thanks for your constructive feedback. Our paper primarily focuses on the problem of **LLM routing**, and therefore the original version of the Introduction centered on the development of LLM routers, with the Related Work section covering representative advances in agent-based systems. We sincerely appreciate the reviewer’s suggestion, which gave us the opportunity to further strengthen and contextualize our related work. In the revised PDF, we added an additional subsection in **Additional Related Work**, where we discuss **Tool-Augmented LLMs and Real-World Agent Ecosystems (lines 888–897)** to provide a more comprehensive view of this line of research.
>
> ---
>
> **Q5. Why are other graph encoders (e.g., GAT, GraphTransformer) not compared or ablated against GARNet?**
>
> **Response:** Thanks for your insightful question. In fact, as presented in **Section 4.3 of our paper (especially lines 488–493)**, we have already provided a detailed discussion and empirical comparison between GARNet and representative homogeneous and heterogeneous graph encoders. These results demonstrate the effectiveness of GARNet through clear performance gains. We believe that these comparisons already provide sufficiently representative evidence demonstrating the superiority of GARNet over other graph encoders.
>
> Following the reviewer’s suggestion, we further expand this analysis by conducting additional ablation studies comparing GARNet against two widely used graph encoders, GAT and GraphTransformer. Specifically:
>
> - **GAT:** Replaces GARNet with a graph attention network.
> - **GraphTransformer:** Replaces GARNet with a GraphTransformer.
>
> We report the results in the following table. Across all five scenarios, GraphPlanner equipped with GARNet achieves the strongest overall performance. Relative to **GAT**, GARNet delivers consistent improvements ranging from **2.8% to 6.1%**, with the most substantial gains observed in the WK and Popular scenarios. Compared with the more computationally heavy **GraphTransformer**, GARNet still provides **2.3%–7.6%** relative improvements while maintaining significantly lower architectural and computational overhead. These findings confirm that our tailored graph encoder **GARNet** not only yields the best accuracy across domains but also serves as a lightweight and efficient alternative to transformer-based graph encoders, reinforcing its importance within the overall GraphPlanner framework. We have incorporated the above discussion and experimental results into **Section F of the Appendix** in the revised PDF.
>
> **Table: Ablation on different graph encoders within the method architecture across five scenarios**
>
> | Setting | Math | Code | CS | WK | Popular |
> |---------|------|------|-----|-----|---------|
> | GAT | 0.643 | 0.739 | 0.756 | 0.358 | 0.493 |
> | GraphTransformer | 0.647 | 0.743 | 0.759 | 0.353 | 0.491 |
> | **GraphPlanner** | **0.670** | **0.760** | **0.780** | **0.380** | **0.520** |

---

> > ### Author Response · Authors · 2025-11-21
> > **Response to Reviewer E6qX (3/3)**
> >
> > **Q6. What motivated the use of Longformer embeddings? Was long-context modeling empirically necessary?**
> >
> > **Response:** Thanks for your thoughtful question. We use Longformer embeddings because some tasks, such as LogicGrid, contain queries that are substantially longer than what common embedding models like BERT can process. Moreover, to ensure practical applicability in real-world usage, GraphPlanner is trained as a **single unified router across all tasks**, which requires a consistent embedding mechanism across heterogeneous query types. Longformer therefore provides a feasible way to handle long-context queries while maintaining consistency across tasks. Of course, Longformer is not the only possible solution for long-context embeddings; it is simply one viable choice. Since handling long-context embeddings is not the core problem our paper aims to solve, we did not elaborate too much on this aspect in the main text.
> >
> > ---
> >
> > **Q7. Can the authors clarify what constitutes “existing LLM workflows” in Phase 1 evaluation (Table 1)? Were these synthetic or from real systems?**
> >
> > **Response:**   Thanks for your constructive question. To clarify, the purpose of our Phase-1 experiments (**illustrated in lines 350-360 of the revised PDF**) is to show that GraphPlanner can optimize LLM routing **on top of any agentic workflow that a user predefines**, regardless of its structure. Therefore, without loss of generality, we construct several representative synthetic workflows as illustrative examples. These setups allow us to clearly demonstrate that GraphPlanner provides superior routing performance compared with existing LLM routers across a diverse range of user-specified workflow configurations.

---

> > > ### Comment · Reviewer_E6qX · 2025-11-26
> > >
> > > I thank the authors for their detailed rebuttal and the additional experiments. The new analyses addressing the flexibility for new roles, the graph encoder ablations, and the expanded discussion on tool-call agents resolve several of my earlier concerns. I have updated my rating accordingly.

---

> > > > ### Author Response · Authors · 2025-11-26
> > > > **Thanks for the Reviewer’s Constructive Feedback**
> > > >
> > > > Thank you for your thoughtful and constructive feedback. We are pleased to hear that our responses have addressed all of your concerns. We are committed to incorporating the suggested changes in our revisions to further enhance the manuscript.

---

### Author Response · Authors · 2025-11-28
**Global Response**

We sincerely thank the AC and all reviewers for their careful and constructive feedback. After the rebuttal, **two reviewers raised their overall scores from 4 to 6** (on **Dec 24** and **Dec 26**, respectively). Notably, **the second reviewer also increased their confidence rating from 3 to 4**, indicating that our new experiments and clarifications fully resolved their earlier concerns.  **The third reviewer remained clearly positive throughout the process**.  During the rebuttal, we focused on clarifying the problem setting, adding targeted experiments and ablations,  and improving the paper’s positioning and presentation.  Below, we summarize the main concerns, our responses, and the updated assessments.

| Dimension | Key Concerns | Our Main Actions | Post-Rebuttal Response |
| --- | --- | --- | --- |
| Contribution & Impact | How substantial is the contribution beyond existing LLM routers and agent frameworks? What exactly is new? | Clarified the **agentic routing** setting and emphasized that GraphPlanner jointly uses workflow graphs, history graphs, and RL to optimize multi-role, multi-round workflows. We also made explicit that **GraphPlanner is a LLM routing approach rather than a multi-agent framework**: it focuses on deciding *which* roles and models to invoke, and is conceptually distinct from MAS-style systems that define the entire agent ecosystem. We added a concise comparison with prior router designs and summarized performance gains across tasks. | Reviewers now see GraphPlanner as a **novel and practically useful routing framework**, and recognize its explicit positioning as a **standalone router (not a generic multi-agent framework)** as an independent contribution. |
| Roles & Workflow Design | Do three fixed roles and specific workflows limit generality? Can the router handle new roles? | Introduced **additional roles** (e.g., Thinker, Verifier) and evaluated new-role train / zero-shot / few-shot settings, showing that GraphPlanner can flexibly incorporate new roles while maintaining strong performance. Clarified Depth/Width in workflow construction. | Concerns about flexibility are largely resolved. Reviewers acknowledge that GraphPlanner **generalizes to extended role sets and deeper/wider workflows**, instead of relying on a single hand-crafted design. |
| Graph & History Modeling | Are the custom graph encoder and history graph really necessary versus simpler baselines? | Added **ablations** replacing GARNet with GAT and GraphTransformer, and introduced LLM-based history-summary / history-retrieval baselines. Clarified graph construction (role hubs, history/workflow fusion) in text and figures. | Results show that our graph-based history modeling and GARNet encoder **consistently outperform** these alternatives, which reviewers agree better justifies our modeling choices. |
| Efficiency & Practicality | Need clearer evidence on training cost, inference latency, and feasibility in real systems. | Reported detailed **time-cost comparisons** (data collection, NN training, per-query latency) across routers and explained how parallel rollouts keep training efficient. Expanded discussion of deployment with tool-augmented LLMs and agent platforms. | Reviewers agree that GraphPlanner is **computationally efficient and practically deployable**, with competitive or lower cost than existing routers at higher quality. |
| Evaluation Scope & Presentation | Desire for harder tasks, more concrete examples, and clearer exposition. | Added experiments on **more challenging benchmarks**, new qualitative workflow case studies, and improved figures/text for Phase-1/Phase-2, graph construction, and role behavior. | Reviewers find the **expanded evaluation and clearer presentation** convincing and easier to follow, noting that earlier ambiguities are substantially resolved. |

### Overall Assessment After Rebuttal

- The new experiments and clarifications collectively address the main **methodological, generalization, and efficiency** concerns.
- Both **Reviewer E6qX** and **Reviewer 3pJs** report that their major issues have been fully resolved and  **increase their overall scores from 4 to 6**. Notably, **Reviewer 3pJs also raised the confidence from 3 to 4**,  indicating an even stronger assurance in recommending acceptance of our paper.
- The third reviewer remains positive and supportive after the revision.
- Overall, the reviewers’ post-rebuttal comments indicate a **clear improvement in perceived clarity, robustness, and impact** of *GraphPlanner*, as well as a better understanding of its role as a **dedicated routing component** within larger agentic systems.

### Appreciation

We are deeply grateful to the AC and all reviewers for their thoughtful comments and constructive suggestions. Their feedback has substantially improved the clarity, rigor, and scope of *GraphPlanner: Graph-Based Agentic Routing for LLMs*, and we sincerely appreciate the time and effort they devoted to the review process.

---

### Meta-Review · Area_Chair_Q68T · 2026-01-05

**Summary:**

This paper proposes GraphPlanner, a graph-based agentic router. It formulates the routing problem as a Markov Decision Process (MDP) and leverages a tailored graph encoder, GARNet, together with reinforcement learning to jointly optimize task performance and computational efficiency.

The reviewers raised several concerns, including: (1) the limited set of LLM roles, (2) comparisons with simple graph encoders such as GAT and GraphTransformer, (3) distinctions from related works such as agent workflow generation systems and tool-based LLM ecosystems, (4) the time cost of GraphPlanner, (5) different methods for utilizing historical information, and (6) the lack of challenging agentic benchmarks. These concerns were largely addressed during the rebuttal through additional experiments and clarifications.

Two reviewers explicitly stated that they increased their scores after the rebuttal, while one reviewer was positive from the initial evaluation. Overall, both the reviewers and I believe that this work makes a contribution to agentic LLM routing, and we support its acceptance.

**Reviewer Concerns:**

The reviewers raised several concerns, including (1) the limited set of LLM roles, (2) comparisons with the simple graph encoders (GAT and GraphTransformer), (3) the distinction from the works such as agent workflow generation systems and tool-based LLM ecosystems, (4) the time cost for GraphPlanner (5) the different methods to utilize the historical information (6) the lack of challenging agentic benchmarking. These concerns were largely addressed after the rebuttal through additional experiments and clarifications.

One issue remains partially unresolved: the limited empirical validation in real-world agentic tasks. Although the authors discussed this limitation, it was not demonstrated empirically.

**Reviewer Scores:**

The reviewers E6qX and 3pJs responded to the authors’ rebuttal, and said that most of their concerns had been addressed and updated their scores accordingly. For Reviewer SiqR, the authors further strengthened the paper by adding results on a challenging benchmark (AIME) in response to the reviewer’s questions. Reviewer SiqR was positive in the initial evaluation.

---

### Decision · Program_Chairs · 2026-01-26

Accept (Poster)